# Dynamin-dependent entry of *Chlamydia trachomatis* is sequentially regulated by the effectors TarP and TmeA

Matthew D. Romero [1] & Rey A. Carabeo [1] ✉

*Chlamydia* invasion of epithelial cells is a pathogen-driven process involving two functionally distinct effectors – TarP and TmeA. They collaborate to promote robust actin dynamics at sites of entry. Here, we extend studies on the molecular mechanism of invasion by implicating the host GTPase dynamin 2 (Dyn2) in the completion of pathogen uptake. Importantly, Dyn2 function is modulated by TarP and TmeA at the levels of recruitment and activation through oligomerization, respectively. TarP-dependent recruitment requires phosphatidylinositol 3-kinase and the small GTPase Rac1, while TmeA has a post-recruitment role related to Dyn2 oligomerization. This is based on the rescue of invasion duration and efficiency in the absence of TmeA by the Dyn2 oligomer-stabilizing small molecule activator Ryngo 1-23. Notably, Dyn2 also regulated turnover of TarP- and TmeA-associated actin networks, with disrupted Dyn2 function resulting in aberrant turnover dynamics, thus establishing the interdependent functional relationship between Dyn2 and the effectors TarP and TmeA.

*Chlamydia trachomatis* is an obligate intracellular bacterium that infects ocular and genital epithelial cells, causing potentially irreversible damage to infected individuals[1]. *Chlamydia* features a biphasic developmental cycle divided between metabolically quiescent elementary bodies (EBs) which invade host cells and vegetative reticulate bodies (RBs) which replicate inside membrane vacuoles termed inclusions[2]. Given its obligate intracellular nature, entry into host cells is essential for pathogen survival; consequently, *Chlamydia* possesses a robust suite of resources that regulate its uptake. Invasion also underpins pathogenicity, as it promotes access to the intracellular niche where it hijacks several host cell processes. Initial interaction with host epithelial cells is mediated by a reversible electrostatic interaction between a *Chlamydia* adhesin and host heparin sulfate proteoglycans[3]. Subsequently, *Chlamydia* engages multiple host receptors and delivers a variety of protein effectors via a type III secretion system[4–6]. Signaling from the effectors TarP and TmeA establishes a robust actin modulatory network that induces the assembly of actin-rich structures that engulf invading bacteria[7–9]. The resultant actin recruitment is characteristically highly localized to

invading EBs and exhibits rapid kinetics of actin recruitment and turnover, wherein actin network assembly and disassembly occur within 200 s[7,10]. The majority of studies regarding chlamydial invasion focus on regulation of actin recruitment, while the process of disassembly at the end of invasion remains understudied, despite evidence pointing to its importance to EB uptake. We recently reported that altering the dynamics of actin turnover correlated with decreased invasion efficiency[7].

Actin remodeling downstream of N-WASP and WAVE2 signaling contributes to invasion by forming various actin-rich structures (e.g., hypertrophic microvilli, cap-like pockets, and pedestal-like structures) at sites of *Chlamydia* adhesion, indicating the presence of multiple pathways for pathogen entry[11–13]. Although multiple uptake mechanisms have been implicated as potential pathways for *C. trachomatis* invasion, the role of host dynamins during this process has been controversial. Dynamins are large GTPases that form oligomeric structures in a helical configuration around membrane lipids during clathrin- and caveolin-mediated endocytosis, mediating the scission of vesiculated cargoes following GTP hydrolysis[14]. They are comprised of

[1]Department of Pathology, Microbiology, and Immunology, College of Medicine, University of Nebraska Medical Center, Omaha, NE, USA.
✉e-mail: rey.carabeo@unmc.edu

a catalytic G domain, a lipid-binding pleckstrin homology (PH) domain, and a proline-rich domain (PRD) that interacts with Src homology 3 (SH3) domain-containing proteins[15]. Absent activation, dynamins possess low intrinsic GTPase activity and assemble into dimers or tetramers[16]. These are utilized to generate higher-order oligomers such as half-rings, rings, and helices, the latter forming at the collar of invaginating vesicles[17]. GTP hydrolysis induces a conformational change along the oligomer, promoting constriction followed by vesicle scission. This prompts the rapid turnover of the dynamin superstructure[18]. Dynamin oligomerization is promoted by several activators, including SH3 domain-containing proteins[19], actin filaments[20], and membrane lipids[21]. Many known activators of dynamin oligomerization are present at *C. trachomatis* invasion sites, raising the possibility that dynamin-dependent scission is utilized during the terminal stages of this process. Several host proteins present during invasion are also directly or indirectly targeted by chlamydial effectors[4,22–24], highlighting the level of control the pathogen exerts on the invasion process.

Previous work by several groups reported conflicting results regarding dynamin involvement during invasion. RNA interference of dynamin 2 (Dyn2) restricted *C. trachomatis* uptake[12], bolstering support for a dynamin-dependent uptake mechanism. In contrast, pre-treatment with the dynamin inhibitor MiTMAB, which targets the PH domain of dynamin, did not alter *C. trachomatis* invasion efficiency[11]. However, this study also identified that SNX9, a BAR-domain protein which promotes dynamin oligomerization, is recruited during the invasion and that its depletion attenuated *Chlamydia* entry. Furthermore, overexpression of dominant negative GTPase-inactive Dyn1 K44A did not prevent *C. trachomatis* infection of HeLa cells[25]. Notably, this study did not investigate *C. trachomatis* uptake frequency and did not target Dyn2, the predominant dynamin species expressed in epithelial cells. In this study, we aim to reconcile the controversial involvement of host dynamins during *C. trachomatis* entry, monitoring its involvement using a series of high-resolution tools previously employed to characterize the regulation of actin remodeling during invasion[7].

Given that dynamin interacts both with actin itself and with several proteins that regulate actin polymerization[20,26–29], it has become increasingly apparent that the dynamin GTPase cycle and actin polymerization are co-regulated. On this basis, the secreted chlamydial effectors TarP and TmeA, which are themselves regulators of actin dynamics, likely also regulate host Dyn2 during invasion. Once secreted, TmeA associates with the plasma membrane and activates N-WASP, followed by Arp2/3 complex activation and nucleation of actin polymerization[8,9]. Likewise, TarP signaling activates host signaling proteins such as Rac1, PI3K, and the WAVE2 complex, in addition to recruiting the actin effectors formin and Arp2/3[7,30]. Many host proteins associated with TarP and TmeA signaling are known to regulate Dyn2 oligomerization, such as cortactin[31], EPS8[32], profilin[31,33], and the Arp2/3 complex[34]. Thus, in addition to the previously established role of TarP and TmeA signaling as synergistic effectors of rapid actin kinetics[7,8], it is likely that they coordinate Dyn2 localization dynamics during *Chlamydia* entry.

Here, we demonstrate that Dyn2 is co-recruited alongside actin during *Chlamydia* invasion and mediates efficient engulfment of the pathogen. This phenomenon is contingent upon signaling from both TarP and TmeA, such that TarP signaling promotes local recruitment of Dyn2, whereas TmeA signaling activates Dyn2 potentially by facilitating oligomerization. The application of the Dyn2 activator Ryngo 1-23, which promotes oligomerization and stabilizes Dyn2 polymers, rescues invasion defects associated with TmeA deletion, enhancing its entry efficiency and restoring Dyn2 and actin recruitment dynamics to near wild-type levels. The rescue indicates a Dyn2-related function of TmeA that is linked to oligomerization. Further, we discovered that actin disassembly is dependent on Dyn2 function, thus ensuring the

completion of invasion. Altogether, these findings resolve a long-standing controversy within the field, providing a novel regulatory function of TarP and TmeA in the rapid assembly and disassembly of *Chlamydia* engulfment machinery, and in respectively mediating Dyn2 recruitment and activation. Our findings highlight cooperation between TarP and TmeA to establish their respective actin networks to recruit and activate Dyn2, in addition to their roles in forming engulfment structures.

## Results

### Dynamin 2 and actin are co-recruited during *Chlamydia* entry

Since conflicting reports persist regarding host dynamin 2 (Dyn2) involvement during *C. trachomatis* invasion, we revisited the question and evaluated its recruitment in greater detail using quantitative imaging approaches. We first determined whether Dyn2 was present within entry sites by co-transfecting Cos7 cells with GFP-Dyn2 and iRFP$_{670}$-LifeAct prior to infection with wild-type *C. trachomatis* (MOI = 20) stained with the red fluorescent dye CMTPX. Using live-cell confocal microscopy, we monitored Dyn2 and actin recruitment during entry, acquiring images at 20 s intervals (Fig. 1a). As previously reported[7], we observed rapid actin recruitment, which was concomitant with the arrival of Dyn2 and subsequent rapid uptake of *Chlamydia*, as characterized by the loss of CMTPX-CTL2 signal at the plasma membrane within 200–300 s. In contrast, expression of mutant Dyn2 K44A (Dyn2 DN), which is defective in GTPase binding and hydrolysis and cannot mediate vesicle scission (Fig. 1b), prolonged internalization to 400–700 s. Delayed pathogen uptake following Dyn2 DN expression could arise from several potential sources, such as inefficient Dyn2 recruitment, impaired actin dynamics, or disruptions within the Dyn2 GTPase cycle that prevent vesicle scission. To address each of these possibilities, we employed a previously established protocol for quantitatively assessing host protein recruitment dynamics during *Chlamydia* invasion, starting by characterizing Dyn2 WT and Dyn2 DN recruitment dynamics (Fig. 1c). While both Dyn2 WT and Dyn2 DN were recruited during entry, we noted that Dyn2 DN achieved peak mean fluorescence intensity (MFI) 80 s later than Dyn2 WT and persisted within entry sites for a longer duration, indicating that rapid recruitment of Dyn2 and subsequent rapid entry of *Chlamydia* is contingent upon Dyn2 GTPase activity. To further substantiate this claim, we converted time-lapse images of actin, Dyn2, and *Chlamydia* into kymographs, upon which we indicated the start (i.e., initiation of actin/Dyn2 recruitment) and end (i.e., loss of EB fluorescence) of invasion (Figs. 1d, S1). The duration between initiation of actin/Dyn2 recruitment and pathogen entry was prolonged by expression of Dyn2 DN (Fig. 1d, e), such that *Chlamydia* uptake in cells expressing Dyn2 WT occurred within 180 s, which was delayed by over two-fold (380 s) upon Dyn2 DN expression. Moreover, slow pathogen uptake following Dyn2 DN expression coincided with slower Dyn2 recruitment and turnover (Fig. 1f, g), reducing the rate of Dyn2 recruitment by 40 percent and turnover by 60 percent compared to Dyn2 WT. Altogether, these data indicate that Dyn2 is co-recruited alongside actin during *Chlamydia* entry and that Dyn2 GTPase activity is necessary for efficient recruitment dynamics and rapid pathogen entry.

### Dynamin 2 inhibition restricts *Chlamydia* entry and actin turnover

The recruitment of Dyn2 alongside its role in facilitating rapid pathogen entry suggests that dynamin-dependent uptake is important for *Chlamydia* invasion. Previous reports indicate that Dyn2 self-assembly and actin polymerization are co-regulated[26,29,34], such that delayed *Chlamydia* entry following Dyn2 disruption may be due to defective actin polymerization. To test this, we disrupted Dyn2 activity via pharmacological inhibition or RNA interference before monitoring actin recruitment during *Chlamydia* invasion. Since co-overexpression

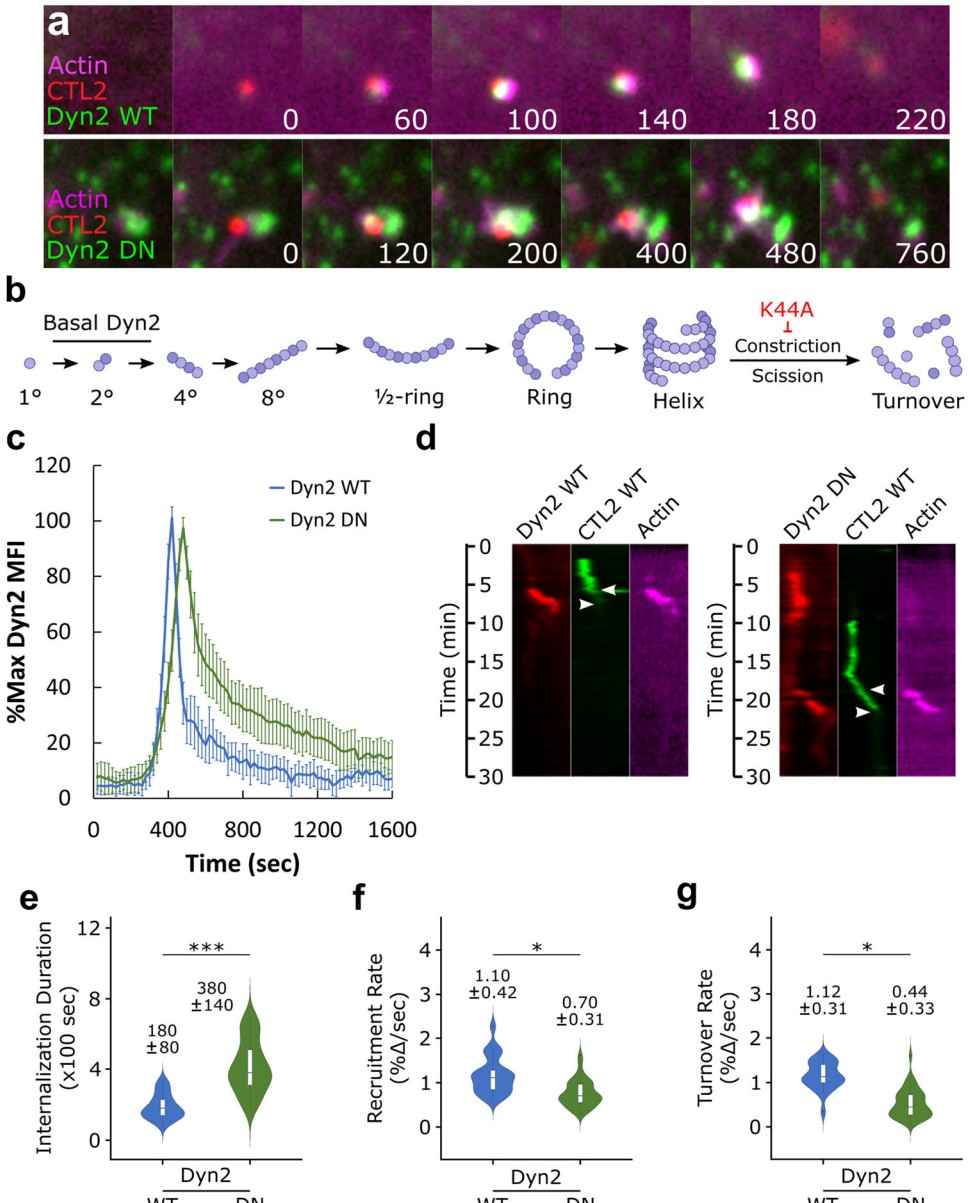

**Fig. 1 | Dynamin 2 and actin are co-recruited during *Chlamydia* entry. a** Cos7 cells were transfected with GFP-Dyn2 WT or K44A (DN) and miRFP-670 LifeAct for 24 h prior to infection with wild-type *Chlamydia* at MOI = 20. Infection was monitored by live-cell confocal microscopy using a Nikon CSU-W1 spinning disk microscope, obtaining images every 20 s for 30 min to identify sites exhibiting actin and Dyn2 co-recruitment. Scale bar = 1 micron. **b** Schematic depicting Dyn2 oligomerization, highlighting disruption of Dyn2 scission by K44A mutation. **c** Mean fluorescence intensity (MFI) of Dyn2 recruitment at *Chlamydia* entry sites was quantified, normalized as percent maximal MFI, and plotted onto a line graph depicting %max Dyn2 MFI ± SEM for each timepoint. Background Dyn2 fluorescence was subtracted prior to normalization, which was performed independently for each Dyn2 WT and DN recruitment event. **d** Kymographs depicting RFP-Dyn2, GFP-*Chlamydia*, and far red actin signal over a 30 min timelapse. Top arrow indicates the initiation of protein recruitment and the bottom arrow indicates the

completion of pathogen entry. **e–g** Detailed analysis of each recruitment event obtained via live-cell imaging, plotting the **e** internalization duration, **f** rate of Dyn2 recruitment, and **g** Dyn2 turnover of each event on a violin plot with inset boxplot reporting the median value and interquartile range for each condition. **e** Internalization duration was quantified by calculating the elapsed time between initiation of protein recruitment and termination of pathogen entry, as detailed in Fig. S1. Individual rates of Dyn2 recruitment (**f**) and turnover (**g**) were calculated by measuring the slope derived from basal Dyn2 MFI to peak MFI for recruitment, and peak Dyn2 MFI to basal MFI for turnover, as detailed in Fig. S1. Data was obtained from a minimum *N* = 23 individual rates per treatment/experimental group. Statistical significance was determined by a two-sided Wilcoxon ranked sum. All data are representative of 3 independent experiments, \*$P \le 0.05$, \*\*$P \le 0.01$, \*\*\*$P \le 0.001$. (**b**) was created with BioRender.com released under a Creative Commons Attribution-NonCommercial-NoDerivs 4.0 International license.

of Dyn2 DN and actin may artificially influence actin dynamics, we instead inhibited endogenous Dyn2 using the dynamin inhibitor Dynasore, which mimics Dyn2 DN by restricting Dyn2 GTPase activity and subsequent scission (Fig. 2a). Furthermore, we were limited to ~50% Dyn2 knockdown via RNA interference (Fig. 2b), as further depletion prevented cell adherence and cell proliferation, rendering these cells unsuitable for further analysis. Nonetheless, we noted that

both 25 μM Dynasore treatment and 50% depletion of Dyn2 attenuated actin dynamics during CTL2 WT invasion (Fig. 2b), with prolonged actin retention within entry sites. Interestingly, actin recruitment kinetics were largely unchanged by Dyn2 disruption, yielding comparable rates across all conditions (Fig. 2c). In contrast, actin turnover was significantly attenuated by both Dynasore treatment and Dyn2 siRNA knockdown, with Dynasore treatment halving the actin turnover

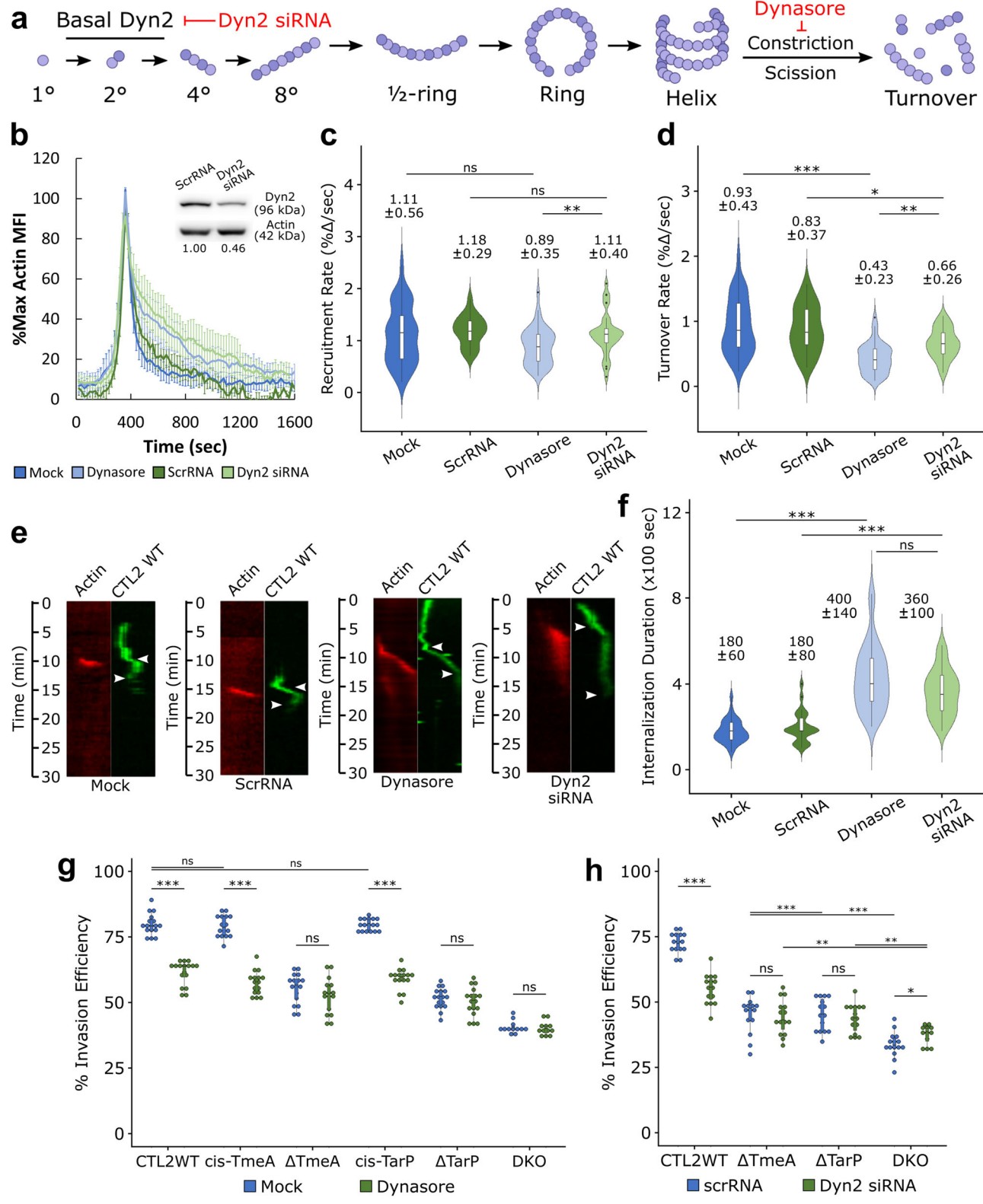

rate, while Dyn2 siRNA treatment slowed actin turnover by 25 percent (Fig. 2d). Given the importance of rapid actin turnover kinetics toward efficient invasion[7], it is possible that Dyn2 inhibition (or its absence) prolongs *Chlamydia* entry through defects in actin turnover. In support, we observed that both inhibition and depletion of Dyn2 delayed *Chlamydia* entry by roughly two-fold (Fig. 2e, f), comparable to the delay observed following Dyn2 DN overexpression (Fig. 1e), indicating

that active Dyn2 is required for efficient actin turnover and rapid *Chlamydia* entry. Moreover, we observed a comparable attenuation in wild-type *Chlamydia* entry efficiency following Dyn2 inhibition (Figs. 2g, S6B) or siRNA depletion (Fig. 2h), reducing *Chlamydia* uptake by roughly 20 percent. Therefore, Dyn2 activity regulates actin turnover during invasion wherein disruption of Dyn2 impedes actin depolymerization within entry sites.

**Fig. 2 | Disruption of Dynamin 2 restricts actin turnover and *Chlamydia* entry.**
**a** Schematic depicting Dyn2 oligomerization, highlighting disruption of
Dyn2 scission by Dynasore treatment. **b** Cos7 cells were transfected with GFP-actin
for 24 h prior to infection with RFP CMTPX-stained wild-type *Chlamydia* (MOI = 20).
Infection was monitored by live-cell confocal microscopy, obtaining images every
20 s for 30 min to identify sites of actin recruitment proximal to invading bacteria.
Actin recruitment at pathogen entry sites was quantified as described earlier
(Fig. 1c) and plotted as %max actin MFI for each timepoint ± SEM compiled from a
minimum $N = 36$ recruitment events per treatment/experimental group. Upon
completion of imaging, cells which received either scramble RNA or Dyn2 siRNA
were lysed in 2× Laemmli buffer, resolving protein expression via Western blot to
determine the knockdown efficiency of Dyn2 siRNA compared to actin loading
control. Kinetics of **c** actin recruitment and **d** actin turnover and **f** internalization
duration were obtained using the same methodology described in Fig. 1e–g. Violin
plots contain a minimum $N = 34$ individual events per treatment/experimental
group, reporting the median value and interquartile range. Statistical significance

was determined by Wilcoxon Rank-sum. **e** Kymographs depicting RFP-Dyn2 and
GFP-*Chlamydia* fluorescence over a 30 min timelapse. Top arrow indicates the
initiation of protein recruitment and the bottom arrow indicates the completion of
pathogen entry. **g, h** HeLa cells were infected with the indicated *Chlamydia* strain at
MOI = 50 and stained using the "in-and-out" method which distinguishes non-
internalized EBs from total cell-associated EBs, as described in "Methods" section.
**g** Cells were pretreated with 25 μM Dynasore for 30 min prior to infection, or
**h** transfected with either scramble or Dyn2-specific siRNA for 24 h prior to infec-
tion. Invasion efficiency of each *Chlamydia* strain is reported as a dotplot with inset
boxplot reporting the median value and interquartile range. Data was collected
from 15 fields per treatment/experimental group, with each field containing an
average of 50 *Chlamydiae*. Statistical significance was determined by pairwise *T*-test
with Bonferroni post-correction. All data are representative of at least 3 indepen-
dent experiments, *$P \leq 0.05$, **$P \leq 0.01$, ***$P \leq 0.001$. (**a**) was created with BioR-
ender.com and released under a Creative Commons Attribution-NonCommercial-
NoDerivs 4.0 International license.

## Signaling from both TarP and TmeA is required for dynamin-dependent entry

Several studies have indicated that mutant *Chlamydia* strains harbor-
ing TarP and TmeA deletion or loss-of-function mutations exhibit
substantially dysregulated pathogen entry[7,35,36]. As such, we monitored
the invasion of *Chlamydia* mutant strains lacking TmeA (ΔTmeA), TarP
(ΔTarP), or both (DKO) to determine if their respective routes of entry
were affected by Dyn2 inhibition or depletion. Loss of either TarP or
TmeA rendered their respective invasion processes resistant to Dyn2
inhibition (Figs. 2g, S6B), likely indicating the utilization of an alter-
native fluid-phase entry mechanism that does not require completion
of the dynamin GTPase cycle (Fig. S2). Entry efficiency of these strains
were similarly insensitive to Dyn2 depletion via RNA interference
(Fig. 2h). Finally, we noted that *cis*-complementation of the ΔTarP and
ΔTmeA mutants (*cis*-TmeA, *cis*-TarP) restored Dynasore sensitiv-
ity (Fig. 2g).

The ΔTarP mutant assembles structures typically associated with
fluid-phase uptake, such as large blooms and mini-ruffles[37] (Fig. S2A, E,
Video S7), potentially accounting for Dyn2 dispensability. Membrane
ruffles are generally associated with fluid-phase uptake, which several
reports described as being dynamin-independent[38–40]. Indeed, ΔTarP
EBs frequently colocalized with the fluid-phase marker Dextran-Alexa
Fluor 647; 40 percent of EBs were dextran positive within 20 min post-
entry (Fig. S2G, H). In contrast, the ΔTmeA mutant retained punctate
recruitment of actin characteristic of wild-type EBs (Fig. S2F, Videos S2,
3, 6, 7) and exhibited a lower incidence of dextran colocalization
(Fig. S2G). Thus, invasion of ΔTmeA EBs is mechanistically distinct
from ΔTarP, adopting a spatial configuration that confers Dyn2-
dependence to the invasion process. As such, the apparent insensi-
tivity of ΔTmeA EBs toward Dyn2 inhibition might reflect that Dyn2 is
required for entry, but present in a non-functional state refractory to
inhibition by Dynasore, which will be addressed in detail later in this
study. Altogether, our data unequivocally reveal that dynamin-
dependent uptake is an important component of *C. trachomatis*
invasion contingent upon both TarP and TmeA signaling, wherein each
effector likely regulates different phases of Dyn2 function.

## TarP and TmeA mediate recruitment and post-recruitment activation of Dyn2, respectively

Strikingly, TarP deletion prevented localized recruitment of Dyn2 at
sites of pathogen entry (Fig. S2, Video S1,7), as defined by the lack of
protein recruitment immediately within sites of EB contact, indicating
that TarP signaling regulates early aspects of Dyn2 recruitment. We
hypothesize that the actin network induced by TarP, rather than TarP
itself, is responsible for Dyn2 recruitment, given actin and Dyn2
colocalization (Fig. 1) and the reported functional relationship
between these proteins[20,29]. To provide a mechanistic basis for TarP-
dependent regulation of Dyn2, we investigated how Dyn2 localization

dynamics are affected by ablation of PI3K/Rac1 signaling, which con-
tributes to TarP-mediated actin remodeling[30] (Fig. 3b). We monitored
Dyn2 recruitment following administration of the Rac-specific inhi-
bitor EHop-016 (10 μM) at entry sites of wild-type and ΔTmeA EBs since
both strains retain TarP signaling (Fig. 3). Rac inhibition did not affect
the rate of Dyn2 recruitment (Fig. 3g) but substantially attenuated its
turnover (Fig. 3h), resulting in prolonged retention of Dyn2 within
CTL2 WT entry sites (Mock = 260 s, EHop = 520 s) (Fig. 3a). Moreover,
Dyn2 recruitment intensity was significantly diminished by Rac inhi-
bition (Fig. S5, Video S4), and ΔTarP mutants were unaffected by EHop-
016 treatment (Fig. S4). Thus, TarP-mediated actin remodeling not
only coordinates local recruitment of Dyn2 within entry sites but also
regulates its retention via Rac1 signaling. Interestingly, Dyn2 localiza-
tion dynamics of ΔTmeA mutants were unaffected by Rac inhibition
(Fig. 3c), exhibiting similar recruitment and turnover rates between
mock- and EHop-treated samples (Fig. 3g, h). TmeA-dependent sensi-
tivity of Dyn2 localization dynamics toward Rac signaling hints at a
significant role for TmeA in Dyn2 function, which likely manifests at
later (i.e., post-recruitment) stages. A previous report that took
advantage of allelic exchange mutagenesis to create *C. trachomatis*
Δ*tarP* mutant strains complemented with TarP versions lacking spe-
cific domains implicated the phosphodomain in invasion[36]. This
observation is consistent with a role for Rac1, which signals from the
phosphodomain. At this time, we cannot conclude if inefficient inva-
sion upon Rac1 inhibition is due to aberrant Dyn2 or actin dynamics at
invasion sites, or both.

We next tested the role of PI3K/Vav2 signaling, which is one of the
Rac-activating pathways linked to TarP, the other being Abi1/Eps8/
Sos1 signaling[30] (Fig. 4d). To determine the functional outcome of PI3K
signaling toward Dyn2 regulation, we monitored invasion of wild-type
and ΔTmeA EBs in the presence of the PI3K inhibitor Wortmannin
(100 nM). Pretreatment with Wortmannin yielded intense and long-
lasting Dyn2 localization relative to mock at wild-type entry sites
(Fig. 4a, b, Fig. S5, Video S5) and attenuated the rate of Dyn2 turnover
(Fig. 4h), consistent with PI3K signaling through Rac (Fig. 3c, h). This
effect is specific to TarP signaling, as PI3K inhibition had no impact on
ΔTarP entry (Fig. S4). Interestingly, PI3K inhibition did not alter Dyn2
recruitment during ΔTmeA invasion (Fig. 4a, b), indicating that in the
absence of TmeA, Dyn2 is not in its proper context to be affected
further by wortmannin treatment. Moreover, wortmannin pretreat-
ment did not alter the invasion efficiency of any strain tested (Fig. 4c)
yet induced a significant delay in CTL2 WT uptake (Mock = 180 s,
Wort = 320 s) (Fig. 4e, f). This disparity may arise due to the enhanced
sensitivity of our kymograph-based internalization assay (Fig. 4e, f),
which employs quantitative fluorescence-based live-cell imaging to
identify invasion defects. The former internalization assay (Fig. 4c)
relies on antibody accessibility to measure invasion efficiency, a low-
resolution approach with the likelihood of missing additional steps

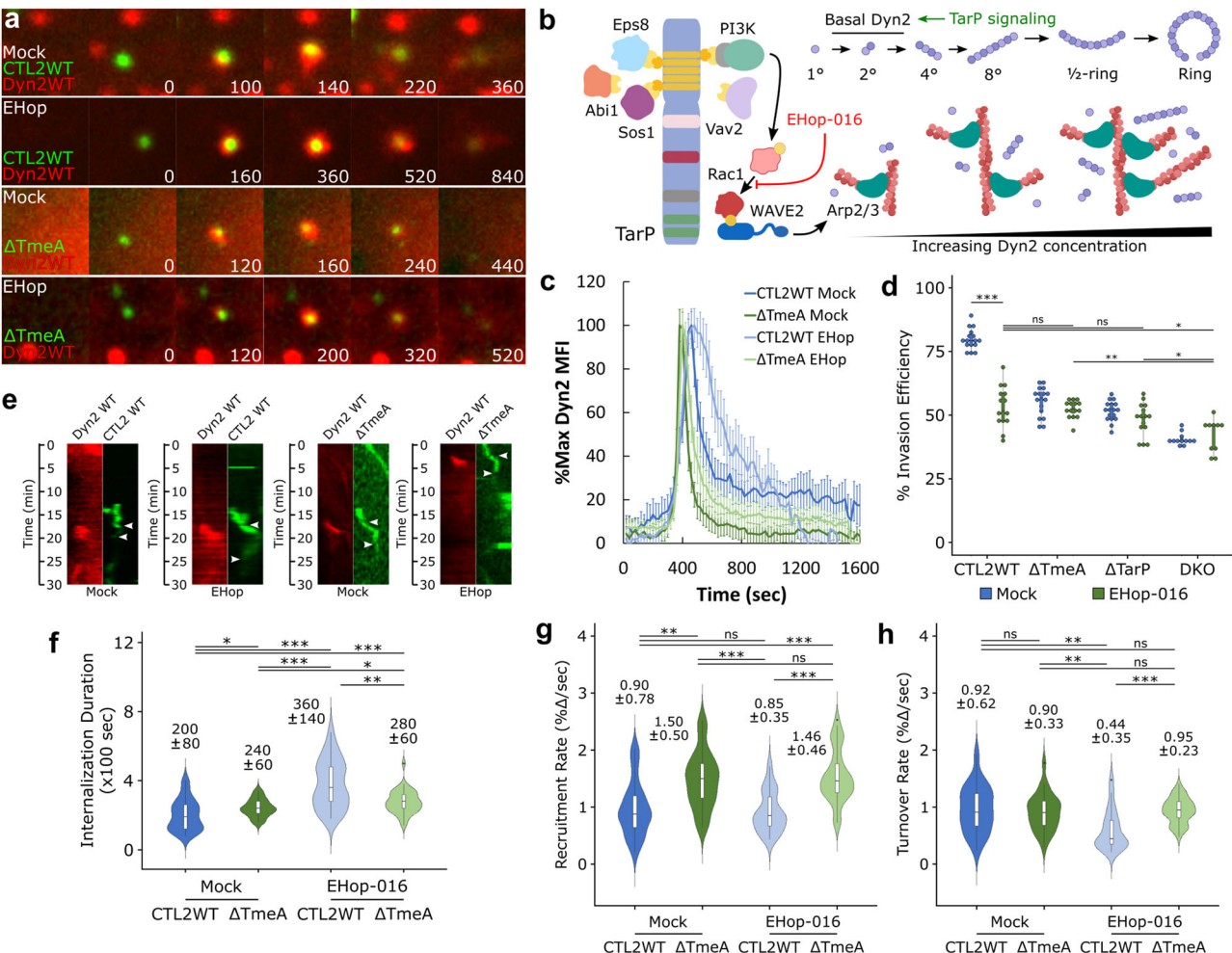

**Fig. 3 | TarP signaling regulates Dynamin 2 recruitment and retention via Rac1 activation.** **a** Cos7 cells were transfected with GFP- or RFP-Dyn2 WT for 24 h prior to infection with wild-type or ΔTmeA EBs at MOI = 20 in the presence or absence of 10 μM EHop-016. Infection was monitored by live-cell confocal microscopy using a Nikon CSU-W1 spinning disk microscope, obtaining images every 20 s for 30 min, and identifying sites exhibiting Dyn2 recruitment during *Chlamydia* entry. Scale bar = 1 micron. **b** Schematic depicting TarP signaling via PI3K/Rac1, subsequent recruitment of actin and Dyn2, and Dyn2 oligomerization, highlighting EHop-016 inhibition of Rac1 and promotion of Dyn2 recruitment by TarP (**c**) Dyn2 recruitment was quantified as described earlier (Fig. 1c) and plotted as %max Dyn2 MFI for each timepoint ± SEM compiled from a minimum *N* = 20 recruitment events per treatment/experimental group. **d** HeLa cells were treated with 10 μM EHop-016 for 30 min before infection with the indicated *Chlamydia* strains at MOI = 50 and stained using the "in-and-out" method described earlier to quantify pathogen entry efficiency. Invasion efficiency is reported as a dotplot with an inset boxplot reporting the median value and interquartile range. Statistical significance was determined by pairwise *T*-test with Bonferroni post-correction. **e** Kymographs depicting RFP-Dyn2 and GFP-*Chlamydia* fluorescence over a 30 min timelapse. Top arrow indicates the initiation of protein recruitment and the bottom arrow indicates the completion of pathogen entry. **f**–**h** Internalization duration (**f**) and kinetics of Dyn2 recruitment (**g**) and turnover (**h**) were obtained using the same methodology described in Fig. 1e–g. Violin plots contain a minimum *N* = 20 individual events per treatment/experimental group, reporting the median value and interquartile range. Statistical significance was determined by a two-sided Wilcoxon Rank-sum. All data are representative of at least 3 independent experiments, *P ≤ 0.05, **P ≤ 0.01, ***P ≤ 0.001. (**b**) was created with BioRender.com released under a Creative Commons Attribution-NonCommercial-NoDerivs 4.0 International license.

during invasion, such as vesicle closure. In summary, our data indicate that TarP signaling is essential for dynamin-dependent entry of *Chlamydia* and is required for the following – (i) local recruitment of Dyn2 within entry sites, and (ii) regulating its retention as a consequence of the actin network generated via the PI3K/Rac1 signaling axis. The latter could be involved potentially in meeting the threshold concentration of Dyn2 needed for subsequent steps, including oligomerization, constriction, and scission.

Although ΔTmeA EBs recruit Dyn2 in a highly localized and punctate manner, similar to CTL2 WT (Figs. 3a, 4a), inhibition of function via ectopic expression of dominant negative Dyn2 or 25 μM Dynasore treatment did not alter uptake duration or Dyn2 dynamics associated with ΔTmeA (Fig. S3). The apparent insensitivity of the ΔTmeA mutant toward Dyn2 disruption may reflect the presence of a Dyn2-independent ΔTmeA entry mechanism, or that TmeA deletion

prevents key steps of Dyn2 functionality post-recruitment. To distinguish between these possibilities, we employed the Dyn2 activator Ryngo 1-23, a small molecule compound that stimulates Dyn2 oligomerization in a manner comparable to short actin filaments[41]. Accordingly, we quantified *Chlamydia* entry after 30 min preincubation with 40 μM Ryngo 1-23, wherein ΔTmeA invasion efficiency was improved to near wild-type levels (Mock CTL2 WT = 79.8%, Ryngo ΔTmeA = 71.0%) (Figs. 5a, S6D). Moreover, this compound restored normal Dyn2 recruitment dynamics during ΔTmeA entry (Fig. 5b), generating a Dyn2 recruitment profile comparable to mock-treated CTL2 WT (Fig. 5d, g, h). Likewise, both mock CTL2 WT and Ryngo ΔTmeA were internalized within 180 sec, in contrast to mock-treated ΔTmeA (240 sec) (Fig. 5e, f), and that compound-assisted entry reduced the incidence of fluid-phase uptake (Fig. S2F). Taken together, these data suggest that in the absence of TmeA signaling, post-

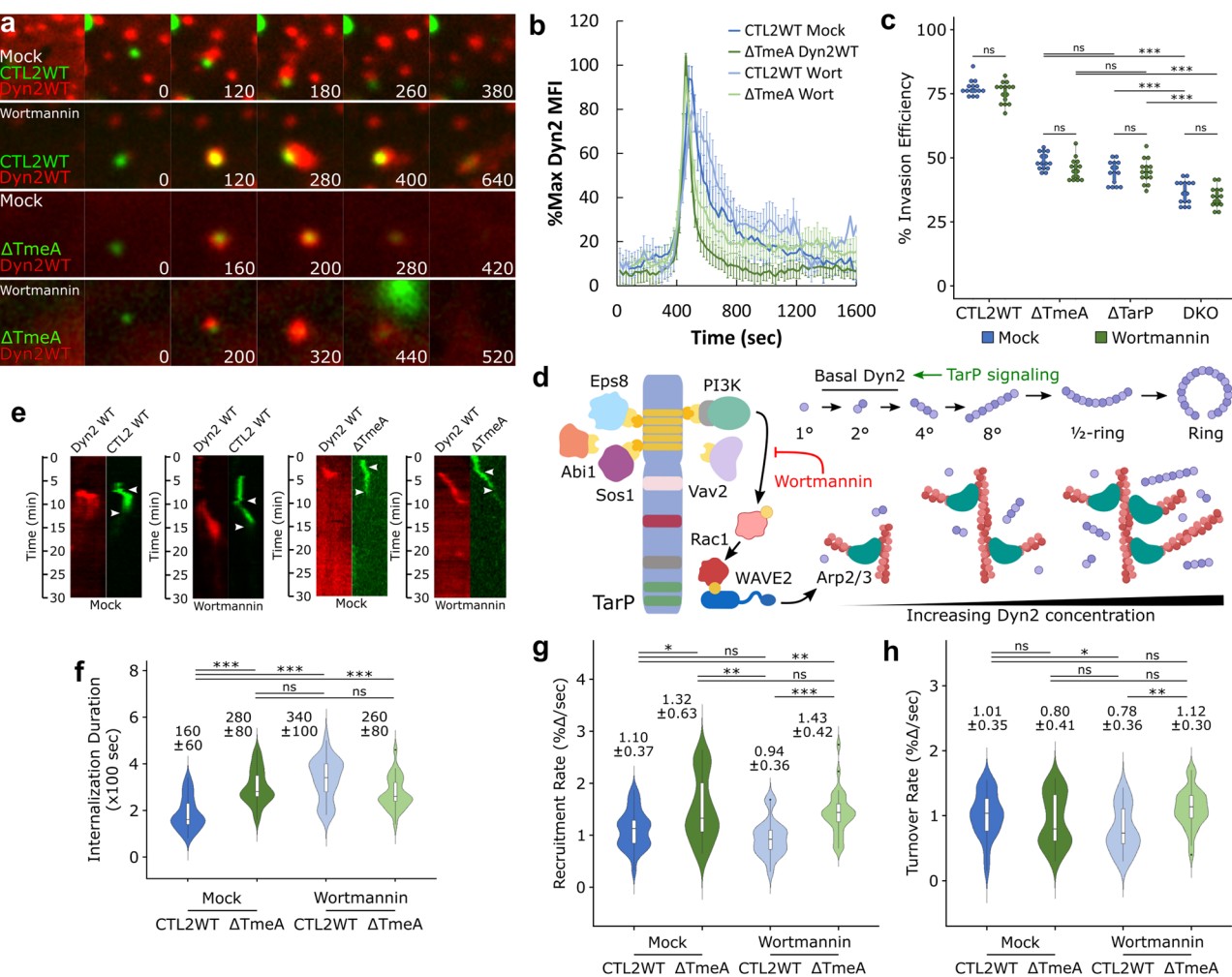

**Fig. 4 | TarP signaling through PI3K/Rac1 governs retention of Dynamin 2.**
**a** Cos7 cells were transfected with GFP- or RFP-Dyn2 WT for 24 h prior to infection with wild-type or ΔTmeA EBs at MOI = 20 in the presence or absence of 40 nM Wortmannin. Infection was monitored by live-cell confocal microscopy using a Nikon CSU-W1 spinning disk microscope, obtaining images every 20 s for 30 min identifying sites exhibiting Dyn2 recruitment during *Chlamydia* entry. Scale bar = 1 micron. **b** Dyn2 recruitment was quantified as described earlier (Fig. 1c) and plotted as %max Dyn2 MFI for each timepoint ± SEM compiled from a minimum *N* = 18 recruitment events per treatment/experimental group. **c** HeLa cells were treated with 40 nM Wortmannin for 30 min before infection with the indicated *Chlamydia* strains at MOI = 50 and stained using the "in-and-out" method described earlier to quantify pathogen entry efficiency. Invasion efficiency is reported as a dotplot with an inset boxplot reporting the median value and interquartile range. Statistical significance was determined by pairwise *T*-test with Bonferroni post-correction.

**d** Schematic depicting TarP signaling via PI3K/Rac1, subsequent recruitment of actin and Dyn2, and Dyn2 oligomerization, highlighting Wortmannin inhibition of PI3K and promotion of Dyn2 recruitment by TarP. (**e**) Kymographs depicting RFP-Dyn2 and GFP-*Chlamydia* fluorescence over a 30 min timelapse. Top arrow indicates the initiation of protein recruitment and the bottom arrow indicates the completion of pathogen entry. **f**–**h** Internalization duration (**f**) and kinetics of Dyn2 recruitment (**g**) and turnover (**h**) were obtained using the same methodology described in Fig. 1e–g. Violin plots contain a minimum *N* = 20 individual events, reporting the median value and interquartile range. Statistical significance was determined by a two-sided Wilcoxon Rank-sum. All data are representative of at least 3 independent experiments, *$*P ≤ 0.05$, **$P ≤ 0.01$, ***$P ≤ 0.001$. (**d**) was created with BioRender.com released under a Creative Commons Attribution-NonCommercial-NoDerivs 4.0 International license.

recruitment activation of Dyn2 does not occur. Given that Ryngo activation of Dyn2 proceeds via its ability to promote oligomerization, its rescue of the ΔTmeA mutant suggests an oligomerization-related role for TmeA. In contrast, invasion efficiency, Dyn2 localization dynamics, and duration of internalization associated with wild-type CTL2 were all negatively affected by Ryngo (Fig. 5a–f). A possible explanation may be that joint activation of Dyn2 by both TmeA signaling and Ryngo administration induces Dyn2 hyperactivation that prevents normal completion of the Dyn2 GTPase cycle. Indeed, Gu et al. found that Ryngo 1-23 abrogated Dyn1 helical collar assembly, instead promoting stacked ring assembly (Fig. 5c). They reported reduced GTPase activity in and attenuated vesicle scission by stacked rings compared to helices[41]. Additionally, CTL2 WT entry was comparably attenuated by either Dynasore-mediated inhibition of Dyn2

(Fig. 2g) or Ryngo-mediated Dyn2 activation (Fig. 5a), implying that dynamin-dependent entry of *Chlamydia* is sensitive to both hypo- and hyperactivation of Dyn2. In contrast, Dyn2 dynamics and function were restored by Ryngo treatment in ΔTmeA EB invasion because Dyn2 proteins could be in a pre-oligomerization state. In summary, these data indicate that TmeA signaling is linked to Dyn2 functionality, potentially at the level of oligomerization based on the previously characterized function of Ryngo.

**Actin turnover is correlated with Dynamin 2 activation status and *Chlamydia* uptake**
Previous studies have identified that TmeA deletion dysregulates the actin network generated by *Chlamydia* during invasion, causing poor actin retention and abnormally fast actin turnover[7–9]. Moreover, in this

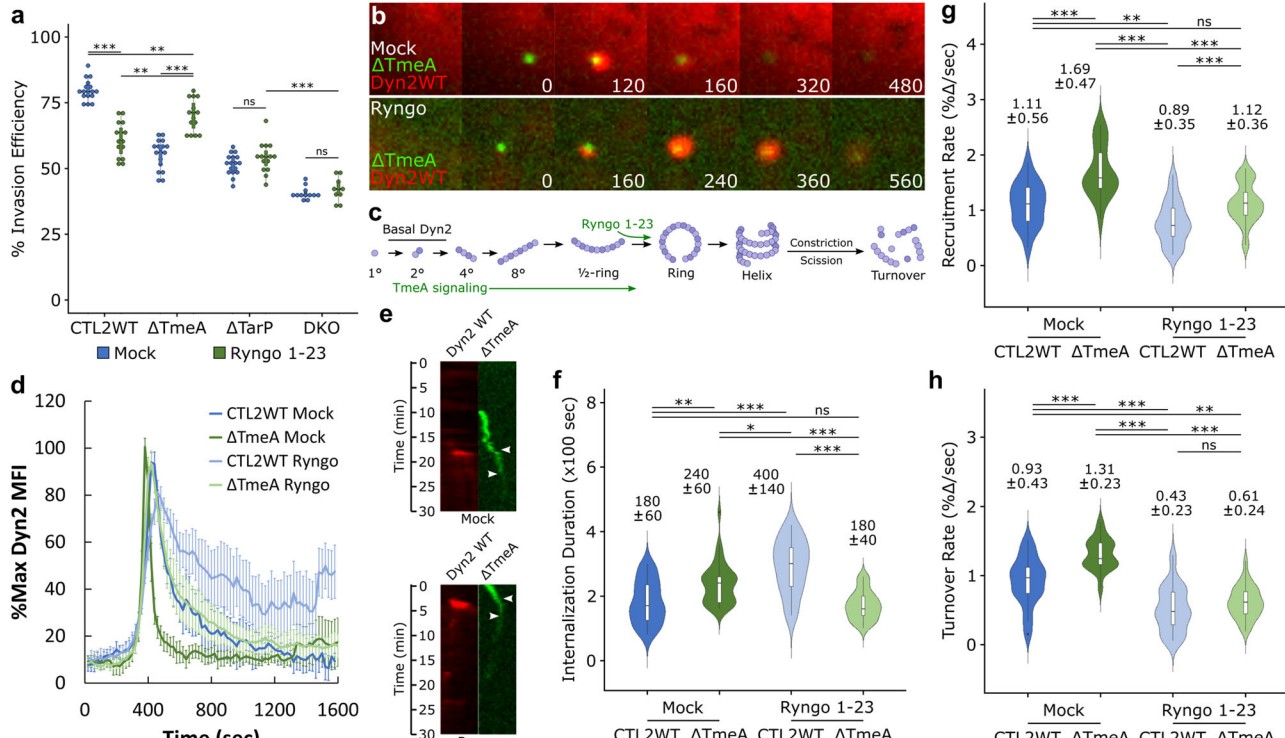

**Fig. 5 | Dynamin 2 activator Ryngo 1-23 rescues invasion defects associated with TmeA deletion. a** HeLa cells were treated with 40 μM Ryngo 1-23 for 30 min before infection with the indicated *Chlamydia* strains at MOI = 50 and stained using the "in-and-out" method described earlier to quantify pathogen entry efficiency. Invasion efficiency is reported as a dotplot with an inset boxplot reporting the median value and interquartile range. Statistical significance was determined by pairwise *T*-test with Bonferroni post-correction. **b** Cos7 cells were transfected with RFP-Dyn2 WT for 24 h prior to infection with ΔTmeA EBs at MOI = 20 in the presence or absence of 40 μM Ryngo 1-23. Infection was monitored by live-cell confocal microscopy using a Nikon CSU-W1 spinning disk microscope, obtaining images every 20 s for 30 min, highlighting Dyn2 recruitment at ΔTmeA entry sites. Scale bar = 1 micron. **c** Schematic depicting Dyn2 oligomerization, promotion of Dyn2 self-assembly by TmeA signaling, and enhancement of Dyn2 ring assembly via Ryngo 1-23 treatment. **d** Dyn2 recruitment was quantified as described earlier (Fig. 1c) and plotted as % max Dyn2 MFI for each timepoint ± SEM compiled from a minimum *N* = 19 recruitment events per treatment/experimental group. **e** Kymographs depicting RFP-Dyn2 and GFP-*Chlamydia* fluorescence over a 30 min timelapse. Top arrow indicates the initiation of protein recruitment and the bottom arrow indicates the completion of pathogen entry. **f**–**h** Internalization duration (**f**) and kinetics of Dyn2 recruitment (**g**) and turnover (**h**) were obtained using the same methodology described in Fig. 1e–g. Violin plots contain a minimum *N* = 19 individual events per treatment/experimental group, reporting the median value and interquartile range. Statistical significance was determined by a two-sided Wilcoxon Rank-sum. All data are representative of at least 3 independent experiments, \*P ≤ 0.05, \*\*P ≤ 0.01, \*\*\*P ≤ 0.001. (**c**) was created with BioRender.com released under a Creative Commons Attribution-NonCommercial-NoDerivs 4.0 International license.

study, we have noted a functional link between Dyn2 activity and actin turnover, wherein actin localized to invasion sites was abnormally persistent upon pharmacological inhibition of Dyn2 or upon expression of Dyn2 K44A (Figs. 2b, S1F), resulting in delayed pathogen uptake. In light of these observations, we opted to evaluate the influence of the dynamin activator Ryngo 1-23 on actin kinetics to determine whether compound-mediated restoration of Dyn2 activity within ΔTmeA entry sites also restores normal actin dynamics. While administration of Ryngo prior to infection strongly increased the persistence of actin recruitment at entry sites of both wild-type and ΔTmeA EBs (Fig. 6a), relative to the respective mock-treated controls, turnover dynamics associated with ΔTmeA EBs were indistinguishable from mock-treated wild-type control (Fig. 6a–c, Videos S2, S3). We also observed that Ryngo treatment restored the internalization duration of ΔTmeA mutants to levels of mock-treated CTL2 WT (Fig. 6e, f). However, when invasion signaling was intact, i.e. when TarP and TmeA are both present, the additional Dyn2 activation by Ryngo had a negative effect on actin turnover and pathogen uptake (Fig. 6a, f, Video S3). This paralleled the effects of Ryngo on Dyn2 recruitment (Fig. 5), underscoring a possible relationship between actin disassembly and Dyn2 turnover (Fig. 6d). Indeed, either insufficient Dyn2 activity (i.e., Dynasore treatment, Dyn2 DN, Mock/ΔTmeA; Fig. 2b–d) or Dyn2 hyperactivation (i.e., Ryngo/CTL2 WT; Fig. 6a–f) results in similar dysregulated actin turnover and delayed pathogen uptake.

Collectively, our data is consistent with a model whereby actin remodeling by TarP and TmeA, in addition to forming engulfment structures, also ensures Dyn2 recruitment and activation. With Dyn2 regulating actin turnover, this self-contained invasion mechanism ensures that disassembly of the invasion structures is properly coordinated with a successful scission event indicated by Dyn2 turnover. Importantly, data obtained from primary cervical epithelial cells (Fig. S6) are consistent with data obtained from immortalized cell lines (Cos7, HeLa), indicating that the mechanism described herein for Dyn2-dependent *Chlamydia* entry is also present in the native host of *C. trachomatis*.

## Discussion

In this study, we conclusively demonstrated that *C. trachomatis* utilizes host Dyn2 to complete invasion. Dyn2 function is modulated by the effectors TarP and TmeA, which respectively mediate Dyn2 recruitment and activation within invasion sites (Fig. 7). Neither TarP nor TmeA mediate direct interaction with Dyn2 to facilitate recruitment and oligomerization[7,8]; instead, TarP and TmeA signaling and their respective actin networks are intertwined with Dyn2 functionality. Interestingly, Dyn2 influences actin turnover, wherein perturbation of Dyn2 function induces persistent actin retention. This functional interdependence constitutes a self-regulating system, such that Dyn2 function and pathogen engulfment are regulated by the actin network

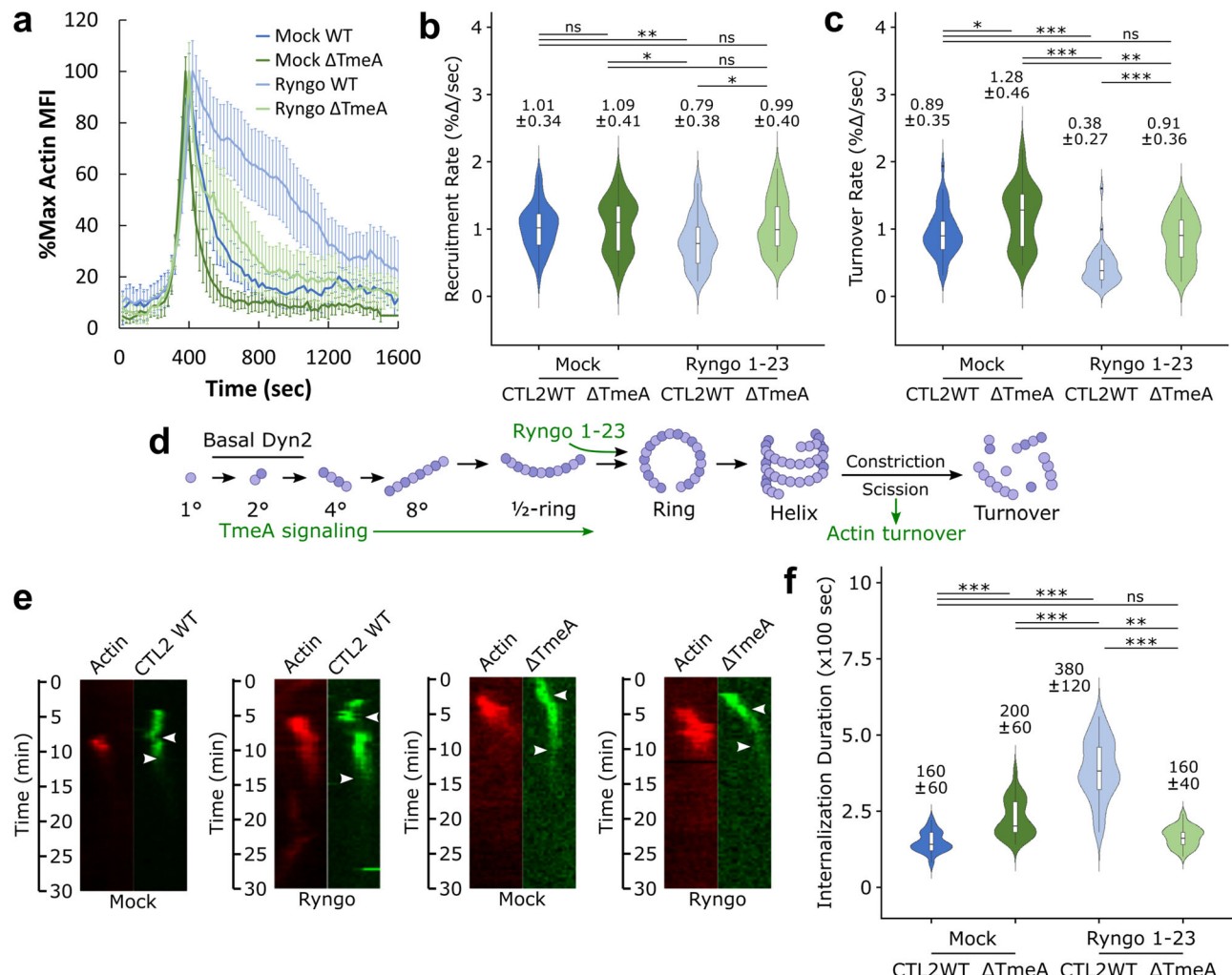

**Fig. 6 | Actin turnover is correlated with Dynamin 2 activation status and *Chlamydia* uptake. a** Cos7 cells were transfected with GFP-Actin or mRuby-LifeAct for 24 h prior to infection with wild-type or ΔTmeA EBs at MOI = 20 in the presence or absence of 40 μM Ryngo 1-23, monitoring pathogen invasion via live-cell con-focal microscopy. Actin recruitment was quantified as described earlier (Fig. 1c) and plotted as %max actin MFI for each timepoint ± SEM compiled from a minimum *N* = 21 recruitment events per treatment/experimental group. **b, c** Kinetics of actin recruitment (**b**) and turnover (**c**) were obtained using the same methodology described in Fig. 1f, g. Violin plots contain a minimum *N* = 21 individual events per treatment/experimental group, reporting the median value and interquartile range. Statistical significance was determined by a two-sided Wilcoxon Rank-sum. (D) Schematic depicting Dyn2 oligomerization, promotion of Dyn2 self-assembly by TmeA signaling, enhancement of Dyn2 ring assembly via Ryngo 1-23 treatment, and

proposed initiation of actin turnover following Dyn2 scission. **e** Kymographs depicting RFP-actin and GFP-*Chlamydia* fluorescence over a 30 min timelapse. Top arrow indicates the initiation of protein recruitment and bottom arrow indicates the completion of pathogen entry. **f** Internalization duration was quantified by calculating the elapsed time between initiation of actin recruitment and termina-tion of pathogen entry, as detailed in Fig. S1. Violin plots contain a minimum *N* = 21 individual events per treatment/experimental group, reporting the median value and interquartile range. Statistical significance was determined by a two-sided Wilcoxon Rank-sum. All data are representative of at least 3 independent experi-ments, *$P \leq 0.05$, **$P \leq 0.01$, ***$P \leq 0.001$. (**d**) was created with BioRender.com released under a Creative Commons Attribution-NonCommercial-NoDerivs 4.0 International license.

assembled via TarP and TmeA signaling. Reciprocally, Dyn2 function and subsequent membrane fission promote actin disassembly and mediate the resolution of engulfment structures. Moreover, TarP and TmeA signaling are sequentially coordinated such that the essential steps of invasion are initiated and completed. Specifically, we found that TarP signaling via PI3K/Rac1 coordinated the initial recruitment and retention of Dyn2 within entry sites, possibly ensuring the pre-sence of Dyn2 at sufficient levels to mediate downstream steps of invasion. Once recruited, Dyn2 is activated by TmeA signaling on the basis that defects associated with TmeA deletion were rescued by administration of the small molecular activator Ryngo 1-23, which promotes Dyn2 oligomerization. Moreover, these data are consistent with previous observations suggesting that TmeA regulates the latter stages of invasion. Finally, our study provides several high-resolution methods for tracking pathogen uptake, enabling detailed analysis of

host-pathogen interactions underpinning *Chlamydia* entry, negating the limitations of an invasion assay dependent on antibody accessi-bility to uninternalized EBs. In summary, we report that Dyn2 activa-tion is an important component of *Chlamydia* invasion, which is regulated synergistically by TarP and TmeA to mediate the scission of *Chlamydia*-containing vesicles and initiate turnover of host proteins following invasion. Altogether, findings underscore the high degree of control *Chlamydia* has over its invasion process.

TarP-deficient strains were incapable of localized and punctate Dyn2 recruitment, indicating that TarP and/or its signaling function is required. Given that Dyn2 directly interacts with several TarP-associated actin regulators, including cortactin[31], EPS8[32], and profilin[31,33], we propose that the actin network generated by TarP sig-naling regulates Dyn2 function. Whether this phenomenon is mediated by direct interaction with actin, which has been reported previously[20],

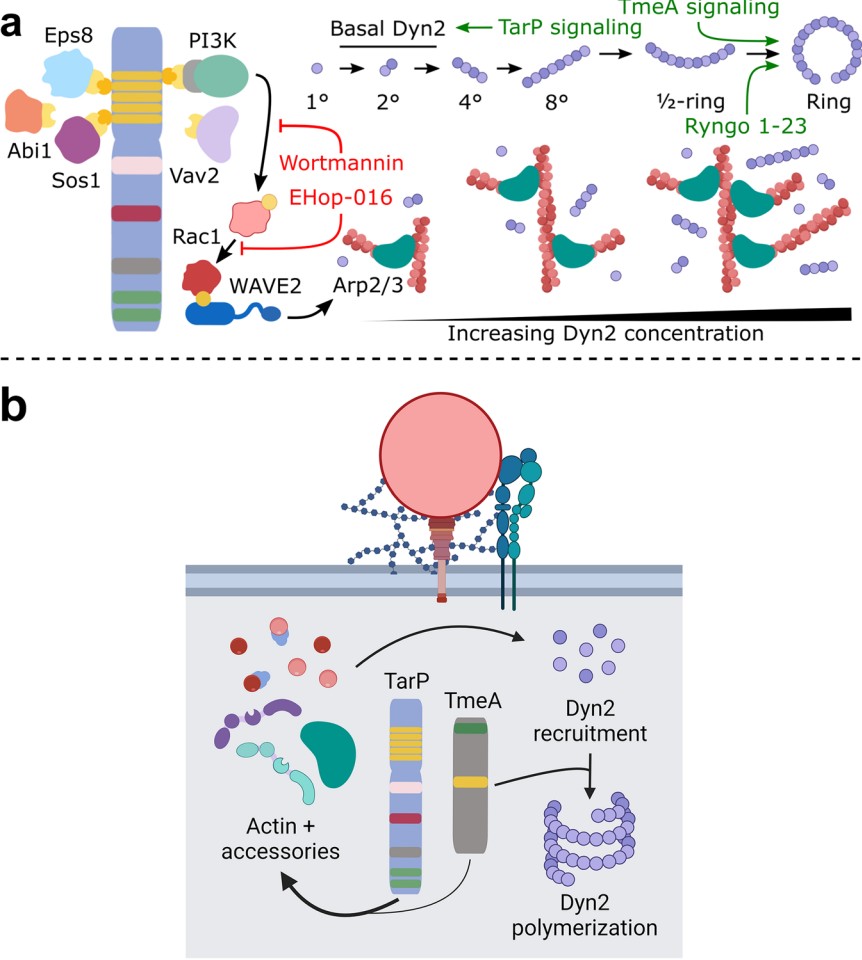

**Fig. 7 | TarP and TmeA sequentially regulate Dyn2 recruitment and activation, respectively. a** TarP signaling promotes Rac1 activation downstream of PI3K signaling within the tandem N-terminal phosphodomains of the effector. Subsequent activation of the Arp2/3 complex enhances actin polymerization and promotes Dyn2 accumulation, providing a basis for TarP-mediated regulation of Dyn2 recruitment that is sensitive to inhibition of PI3K (Wortmannin) or Rac1 (EHop-016). Rescue of TmeA deficiency by the dynamin-activating compound Ryngo 1-23 is consistent with the working model whereby TmeA mediates oligomerization and subsequent assembly of scission-competent Dyn2 helical collars. **b** Signaling for TarP and TmeA promotes dynamin-dependent entry of *Chlamydia* in a sequential and synergetic manner, such that TarP signaling regulates Dyn2 recruitment whereas TmeA signaling regulates Dyn2 polymerization. This figure was created with BioRender.com released under a Creative Commons Attribution-NonCommercial-NoDerivs 4.0 International license.

or by various TarP-associated signaling molecules is not known. One possibility is that TarP-mediated actin remodeling reconfigures the local environment to enrich and retain Dyn2 at sufficient quantities to achieve functionality. For example, robust actin polymerization promotes membrane curvature and supports binding of Bin/amphiphysin/Rvs (BAR) domain proteins, some of which (e.g., SNX9) are known Dyn2 interactors[42]. This would also account for the temporal regulation of Dyn2, wherein the timing of host protein recruitment influences both the concentration and activation of Dyn2. Importantly, we noted that wild-type *Chlamydia* was only partially impaired by Dyn2 disruption (Figs. 2g, h, 5a, S6B, D), implying the utilization of dynamin-insensitive modes of entry by this strain. Although our study demonstrates that Dyn2 and actin dynamics are functionally linked, a comprehensive model of Dyn2 involvement in several invasion mechanisms reported for *Chlamydia* requires further characterization[11–13].

Recently, we reported that TarP signaling uniquely recruited host formins[7], which utilize profilin/actin complexes to acquire monomeric actin[43], and are important regulators of actin polymerization during *Chlamydia* entry. Moreover, the Arp2/3 complex is extensively associated with Dyn2 activity[44–46] and collaborates with host formins to enhance actin remodeling during invasion[7]. Robust actin remodeling

provides a mechanism to ensure Dyn2 recruitment at sufficient levels; consequently, the pathways employed by *Chlamydia* to mediate actin nucleator activation are highly relevant points of Dyn2 regulation. For instance, we observed that TarP signaling via the PI3K/Rac1 axis, which regulates actin polymerization during invasion[30], also governed Dyn2 retention within entry sites. There is precedence for Dyn2 intersecting with actin remodeling, insofar as disruption of Dyn2 dysregulates Rac localization and impairs actin dynamics within lamellipodia[47]. Furthermore, Dyn2 localization dynamics during phagocytosis was determined to be related to actin stability[28,29], in that inhibition of Arp2/3 altered actin dynamics, favored net disassembly, prevented scission of phagocytized particles, and increased Dyn2 persistence[48]. Thus, actin network destabilization following Rac inhibition during invasion could have ultimately interfered with Dyn2-mediated scission and subsequent turnover, yielding an abnormally persistent signal.

We also found that TmeA signaling promoted Dyn2 activation, wherein strains lacking TmeA exhibited defective uptake that was rescued by Ryngo 1-23 administration. Several lines of evidence suggest that TmeA regulates Dyn2 via its previously established role in actin remodeling[7–9]. In vitro, assays identified that both short actin filaments and Ryngo 1-23 stimulate Dyn2 ring assembly[20,38]. Collaboration between TarP and TmeA may regulate Dyn2, through

recruitment and oligomerization respectively. Indeed, both TarP and TmeA were necessary for dynamin-dependent entry, as strains lacking either effector were insensitive to Dyn2 disruption. How might the regulatory contributions of TarP and TmeA be distinguished, given the shared importance of their respective actin remodeling functions? For TmeA, the involvement of N-WASP might offer some clues. N-WASP harbors a proline-rich domain (PRD) that binds proteins with Src homology 3 (SH3) domains. The SH3 domain-containing protein SNX9 interacts with dynamin and stimulates Dyn2 oligomerization[42] and is important for *C. trachomatis* invasion[11]. As such, the interaction between N-WASP and SNX9 might explain Dyn2-dependence toward TmeA signaling. Intriguingly, TmeA also bears similarity with the *C. pneumoniae* secreted effector SemD[49,50], which recruits the BAR-domain proteins PACSIN and SNX9 to induce membrane curvature and promote pathogen engulfment. On this basis, TmeA-mediated Dyn2 regulation could manifest via the formation of SNX9/Dyn2 heterodimers, providing a mechanism of Dyn2 modulation distinct from actin remodeling. Therefore, there are at least two molecular interactions uniquely linking TmeA signaling with the Dyn2 function.

While the precise nature of how TmeA signaling modulates the Dyn2 GTPase cycle remains unknown, analysis of Dyn2 mutants may provide insight toward TmeA/Dyn2 regulation. Studies regarding the formation of progressive higher-order dynamin oligomers have benefited from various mutations that affect protein-protein interactions, GTPase activity, conformational changes, etc. Determining the exact mechanism of compound-mediated rescue following TmeA deletion will require elucidating which oligomeric species of Dyn2 is induced by either Ryngo or TmeA signaling. Mutations that prevent dynamin self-assembly (i.e., Dyn1 I670K[51]) or membrane association (i.e., Dyn2 K562E[52]) could be informative toward this end, as these mutants are scission-incompetent and are not rescued by Ryngo[41,53]. Our working model predicts that these mutants are likewise unresponsive to TmeA signaling. Dyn1 K/E exhibits reduced affinity for actin filaments and is partially rescued by Ryngo[41], whereas Dyn2ΔPRD cannot bind SH3 domain-containing proteins and is dominant negative for endocytosis[54]. Studies incorporating these mutants could clarify whether TmeA signaling operates by mediating Dyn2/actin interactions, or by promoting interaction with SH3 domain-containing proteins like SNX9. Using this report as a foundation, future studies could interrogate the effects of each Dyn2 mutant during *Chlamydia* invasion and determine how effector signaling promotes dynamin-dependent entry.

Interestingly, unlike ΔTmeA, invasion of wild-type *Chlamydia* was impaired by Ryngo treatment such that pathogen entry and kinetics of Dyn2 and actin recruitment were defective. One explanation may be that in certain contexts, Ryngo stimulates Dyn2 oligomerization into a scission-incompetent configuration. FRET analysis of dynamin oligomerization found that Ryngo prompted the assembly of stacked Dyn2 rings around membrane tubules[41], representing a lower-order oligomerization state with insufficient GTPase activity to induce membrane scission. As such, independent stimulation of Dyn2 activation by Ryngo and *Chlamydia*/TmeA signaling, when combined could generate a disproportionate quantity of Dyn2 rings rather than the favored helix. Elimination of *Chlamydia*-specific Dyn2 activation (i.e., ΔTmeA) prevented overstimulation, encouraging proper assembly of higher-order, scission-competent Dyn2 helical collars. Meanwhile, whereas Ryngo pretreatment restored local Dyn2 recruitment at ΔTarP entry sites (Figs. S2C, D, S4B–E, Video S1), it failed to prompt rapid engulfment of ΔTarP EBs and had no rescuing effect on its entry efficiency, suggesting that post-recruitment, Dyn2 needs to be primed for activation, i.e., promotion of oligomerization into helical collars by Ryngo.

Finally, our study identified that both Dyn2 and actin turnover were co-regulated. Mechanistically, Dyn2 turnover is intuitive, occurring concomitantly with membrane scission as a function of GTP hydrolysis[17,18]. As such, Dyn2-mediated scission of *Chlamydia*-containing vacuoles may intrinsically prompt Dyn2 turnover while also providing a signal to initiate actin turnover. Interventions which prevent dynamin-mediated membrane fission also accumulate F-actin around tubulated membranes[55,56], whereas scission is consistently associated with actin turnover, sensitizing actin filaments toward cofilin-mediated severing[29,57,58]. Furthermore, given that dynamin extensively interacts with actin-associated proteins[28,58–60], post-scission turnover of actin regulatory machinery alongside Dyn2 may shift actin regulation toward turnover. Importantly, actin polymerization during *Chlamydia* invasion is both intricately regulated and pathogen-directed[23]; consequently, turnover of actin and other invasion-associated host proteins could be regulated distinctly from turnover associated with routine engulfment of cellular cargoes (i.e., growth factors, transferrin). This could require additional factors that fine-tune their function and/or dynamics to accommodate pathogen-mediated uptake mechanisms. As such, further study is required to gain a more comprehensive perspective on host protein turnover post-invasion.

Overall, our findings of Dyn2 modulation by TarP and TmeA fit well with the proposed pathogen-directed invasion model proposed by Byrne and Moulder[61]. While the majority of molecular studies of chlamydial invasion focus on actin recruitment, we demonstrate here that the latter stages are also targeted by TarP and TmeA, highlighting their central function in the invasion, comprising a self-contained signaling module capable of mediating the initial, middle, and end stages of invasion.

## Methods

### Cell and bacterial culture
Green monkey kidney fibroblast-like (Cos7) cells and cervical adenocarcinoma epithelial (HeLa) cells were cultured at 37 °C with 5% atmospheric CO2 in Dulbecco's Modified Eagle Medium (DMEM; Gibco, Thermo Fisher Scientific, Waltham, MA, USA) supplemented with 10 μg/mL gentamicin, 2 mM L-glutamine, and 10% (v/v) filter-sterilized fetal bovine serum (FBS). HeLa and Cos7 cells were cultured for a maximum of 15 passages for all experiments. McCoy B mouse fibroblasts (originally from Dr. Harlan Caldwell, NIH/NIAID) were cultured under comparable conditions. Primary human cervical epithelial cells (HCECs, ATCC PCS-0480-011, Lot 80306190) were cultured at 37 °C with 5% atmospheric CO2 in Cervical Epithelial Cell Basal Medium (CECBM, ATCC PCS-480-032) supplemented with materials obtained within Cervical Epithelial Growth Kit (ATCC PCS-080-042). All cell lines are routinely verified to be mycoplasma-free using the ATCC universal mycoplasma detection kit (ATCC 30-1012K). *Chlamydia trachomatis* serovar L2 (434/Bu) was propagated in McCoy cells and EBs were purified using a Gastrografin density gradient as described previously[62].

### Reagents
Wortmannin (Selleck, Houston, TX, USA) was diluted upon receipt to 40 mM stock concentration in DMSO, Dynasore (Cayman Chemical, Ann Arbor, MI, USA), and EHop-016 (Cayman) were diluted to 25 mM stock concentration in DMSO, and Ryngo 1-23 (Abcam, Cambridge, MA, USA) was diluted to 20 mM stock concentration in DMSO. All inhibitors were dispensed into single-use aliquots and stored at −20 °C for no longer than 1 year after receipt. Wortmannin was diluted to a working concentration of 40 nM (1:10,000), Dynasore was diluted to a working concentration of 25 μM (1:1000), EHop-016 was diluted to a working concentration of 10 μM (1:2500), and Ryngo 1-23 was diluted to a working concentration of 40 μM (1:500), each using supplemented DMEM as diluent.

### Invasion assay
*C. trachomatis* internalization efficiency was conducted using HeLa cells and was performed as described previously[10]. Briefly, HeLa cells

were seeded in 24-well plates containing acid-etched glass coverslips and allowed to adhere overnight. Invasion assays using HCECs were seeded at 50000 cells/well in a 24-well plate and grown until confluent, according to manufacturer directions. Cells were pretreated with Wortmannin (40 nM), Dynasore (25 µM), EHop-016 (10 µM), or Ryngo (40 µM) for 30 min prior to infection. Dyn2 siRNA or scramble RNA were transfected and allowed to incubate 24 h prior to infection. Following inhibitor treatment or RNA interference, cells were infected with EBs derived from wild-type *C. trachomatis* L2 (434/Bu), *C. trachomatis* in which TarP, TmeA, or both were deleted by FRAEM (ΔTarP, ΔTmeA, ΔTmeA/ΔTarP), or *C. trachomatis* in which TarP or TmeA expression was restored by *cis*-complementation (*cis*-TarP, *cis*-TmeA) at MOI = 50. EBs were allowed to attach onto HeLa cells for 30 min at 4 °C before rinsing coverslips with cold HBSS, followed by the addition of supplemented DMEM prewarmed to 37 °C, before incubating cells at 37 °C for 10 min. After incubation, cells were stringently washed with cold HBSS containing 100 µg/mL heparin to remove any transiently adherent EBs before fixation in 4% paraformaldehyde at room temperature for 15 min. Fixed cells were labeled with a mouse monoclonal anti-MOMP antibody (Novus Biologicals, Centennial, CO, USA #NB10066403), rinsed with 1× PBS, and fixed once more in 4% paraformaldehyde for 10 min. Next, cells were permeabilized using 0.1% (w/v) Triton X-100 for 10 min at room temperature, rinsed with HBSS, and labeled with rabbit polyclonal anti-*Chlamydia trachomatis* antibody (Abcam ab252762). Cells were then rinsed in 1× PBS and labeled with Alexa Fluor 594 anti-mouse (ThermoFisher #A11032, Waltham, MA, USA) and Alexa Fluor 488 anti-rabbit (ThermoFisher #A11034) IgG secondary antibodies. Coverslips were mounted and observed on a Nikon CSU-W1 confocal microscope (Nikon, Melville, NY, USA), obtaining Z-stacks using a 0.3-micron step size across the height of the cell monolayer. Monolayer Z-stacks were transformed via Z-projection according to maximal fluorescence intensity in ImageJ prior to quantifying percent invasion efficiency as follows: total EBs (green) – extracellular EBs (red)/total EBs (green) × 100%.

## Quantitative live-cell imaging of *Chlamydia* invasion

Cos7 cells were seeded onto Ibidi µ-Slide 8-well glass-bottomed chambers (Ibidi, Fitchburg, WI, USA) and allowed to adhere overnight prior to transfection. Cells were transfected with fluorescent proteins as indicated, using Lipofectamine 3000 (ThermoFisher, Waltham, MA, USA) according to manufacturer directions. Transfection was allowed to proceed overnight before replacing media with fresh DMEM + 10% FBS/2 mM L-glutamine and allowing protein expression to continue for a total of 24 h post-transfection. Transfection efficiency was verified on a Nikon CSU-W1 spinning disk confocal microscope prior to application of DMEM containing Wortmannin (40 nM), Dynasore (25 µM), EHop-016 (10 µM), or Ryngo (40 µM). For RNA interference, Dyn2 siRNA or scramble RNA was co-transfected alongside GFP-actin or mRuby-LifeAct and allowed to incubate for 24 h prior to imaging. Wells were individually infected with CMTPX-labeled wild-type *C. trachomatis* L2 (434/Bu) unless otherwise indicated, at MOI = 20 and promptly imaged using a 60x objective (NA 1.40) in a heated and humidified enclosure. Images were collected once every 20 s for 30 min, with the focal plane maintained using an infrared auto-focusing system. Upon completion of the imaging protocol, the next well was infected, and imaging repeated; mock-treated wells were imaged first to allow inhibitor treatment sufficient time to achieve inhibition. Images were compiled into videos using NIH ImageJ and analyzed to identify protein recruitment events. The mean fluorescence intensity (MFI) of recruitment events was measured for each timepoint alongside the local background MFI of a concentric region immediately outside the recruitment event. Background MFI was subtracted from recruitment MFI for each timepoint and normalized as percent maximal fluorescence intensity for each timepoint, repeating this normalization process for each recruitment event.

## Fixed-cell Immunofluorescence

Primary HCECs were seeded at 50000 cells/well in a 24-well plate containing acid-etched glass coverslips and grown until confluent. Cells were pretreated with Dynasore (25 µM), Ryngo (40 µM), or DMSO control for 30 min prior to infection. Following inhibitor treatment, cells were infected with wild-type, ΔTmeA, or ΔTarP elementary bodies at MOI = 50. EBs were allowed to attach onto HCECs for 30 min at 4 °C before rinsing coverslips with cold HBSS, followed by the addition of supplemented DMEM containing inhibitor prewarmed to 37 °C, before incubating cells at 37 °C for 10 min. Cells were rinsed in cold HBSS to halt invasion prior to fixation in 4% paraformaldehyde at room temperature for 15 min. HCECs were permeabilized with 0.1% (w/v) Triton X-100 for 10 min at room temperature, rinsed with HBSS, and labeled with mouse monoclonal anti-MOMP (Novus #NB10066403) and rabbit anti-Dyn2 (Thermo PA1-661) antibodies for 1 h at room temperature. Cells were rinsed in 1× PBS and labeled with Alexa Fluor 488 anti-mouse (ThermoFisher # A-11001) and Alexa Fluor 594 anti-rabbit IgG (ThermoFisher # A-11012) secondary antibodies. Coverslips were mounted and observed on a Nikon CSU-W1 confocal microscope (Nikon), obtaining Z-stacks using a 0.2 micron step size across the height of the cell monolayer. Quantification of Dyn2+ and Actin+ EBs was performed by thresholding fluorescent *Chlamydia* signal using the TrackMate plugin. Briefly, the plugin tracks EB signals throughout the Z-stack reporting actin, Dyn2, and Chlamydia mean fluorescence intensity across all Z-stacks. *Chlamydia* colocalization with either actin, Dyn2, or both were defined as events with elevated fluorescence intensity, positive signal-to-noise ratio, and convergence of actin/Dyn2 signal parallel with *Chlamydia* signal across Z-stacks.

## RNA interference

Cos-7 or HeLa cells were seeded onto Ibidi µ-Slide 8-well glass-bottomed chambers (live-cell imaging) or in 24-well plates containing acid-etched glass coverslips (invasion assay) and allowed to adhere overnight. Mission esiRNAs were custom-ordered to target Cos7 Dyn2 mRNA, ensuring that the resultant esiRNA targeted a shared sequence found in all recorded mRNA transcript variants. Cells were transfected with either 100 nM Mission anti-Dyn2 esiRNA (Eupheria Biotech, Dresden, Germany) or 100 nM Trilencer-27 Universal scrambled negative control (Origene SR30004, Rockville, MD, USA) using Lipofectamine RNAiMAX reagent (ThermoFisher) according to manufacturer directions. Incubation was allowed to proceed for 24 h before conducting live-cell imaging or invasion assays using methods described earlier. Lysates for Western blotting were obtained from Cos7 cells by applying 2× Laemmli buffer to cells after the completion of live-cell imaging.

## Western blotting

Lysates generated as described above were resolved via SDS-PAGE in 10% polyacrylamide gels at 120 volts for 1.5 h or until the dye front has begun to evacuate the bottom of the gel cassette. Gels were transferred onto 0.45 µM pore size nitrocellulose in 1× Towbin buffer + 10% methanol at 90 mA for 16 h. Western blots were blocked in 5% bovine serum albumin for 1 h, briefly rinsed in Tris-buffered saline + 0.1% Tween-20 (TBST), and incubated with appropriate primary antibody for 1 h. Blots were then washed three times for 5 min in TBST and probed with appropriate HRP-conjugated secondary antibodies for 1 h. Protein bands were resolved by chemiluminescence using Immobilon Western HRP Substrate (Millipore Sigma, St Louis, MO, USA). Dynamin 2 knockdown efficiency was calculated by densitometry analysis, comparing the ratio of Dyn2 antibody signal (Thermo PA1-661) against β-actin loading control (Abcam ab49900).

## Dextran uptake assay

Cos7 cells were seeded onto 24-well plates containing acid-etched glass coverslips and allowed to adhere overnight. Cells were treated

with 40 µM Ryngo 1-23, 25 µM Dynasore, or DMSO in media containing 100 µg/mL Dextran-Alexa Fluor 647; 10,000 MW (Thermo D22914) and incubated at 37 °C for 30 min. Cells were infected with wild-type or mutant *Chlamydia* strains at MOI = 50, synchronizing infection by sedimentation at 4 °C on a rocking incubator for 30 min. Infection was initiated by the addition of prewarmed media, followed by incubation in a 37 °C incubator for 20 min prior to fixation in 4% paraformaldehyde for 10 min at room temperature. Fluorescent dextran and inhibitor were maintained in media for each indicated stage. Cells were then permeabilized using 0.1% (w/v) Triton X-100 for 10 min at room temperature, rinsed with HBSS, and labeled with a mouse monoclonal anti-MOMP antibody (Novus #NB10066403). Cells were rinsed in 1× PBS and labeled with Alexa Fluor 488 anti-mouse (ThermoFisher # A-11001) IgG secondary antibody. Coverslips were mounted and observed on a Nikon CSU-W1 confocal microscope (Nikon), obtaining Z-stacks using a 0.3-micron step size across the height of the cell monolayer. Monolayer Z-stacks were analyzed using the TrackMate plugin included in FIJI (ImageJ), which quantifies Dextran and *Chlamydia* fluorescence intensity for each *Chlamydia* elementary body across all Z-slices. Dextran+ events were defined as *Chlamydia* particles containing elevated Dyn2 fluorescence intensity and positive signal-to-noise ratio (SNR), where SNR measures the relative intensity of Dyn2 fluorescence within the region of interest compared to the background. Details regarding TrackMate analysis parameters can be found at https://imagej.net/plugins/trackmate/analyzers/. Proportion of Dextran+ events were calculated as follows: [Dextran+ EBs (magenta/green)/Total EBs (green)] × 100%.

## Plasmids and DNA preparation

pEGFP-Actin–C1[63] was a gift from Dr. Scott Grieshaber (University of Idaho), and mRuby-LifeAct-7 (Addgene plasmid #54560) was a gift from Michael Davidson. Dyn2-pmCherryN1 was a gift from Dr. Christien Merrifield (Addgene plasmid #27689), RFP Dynamin 2 K44A was a gift from Dr. Jennifer Lippincott-Schwartz (Addgene plasmid #128153), WT Dyn2 pEGFP was a gift from Dr. Sandra Schmid (Addgene plasmid #34686), and GFP-Dynamin 2 K44A was a gift from Dr. Pietro De Camilli (Addgene plasmid #22301). Upon receipt, bacterial stab cultures were streak-plated onto LB agar containing appropriate antibiotic (Kanamycin, carbenicillin) for each plasmid. Resultant antibiotic-resistant colonies were selected and propagated in LB broth + antibiotic for plasmid isolation prior to sequence verification. All plasmids were isolated using MiniPrep DNA isolation kits (Qiagen, Valencia, CA, USA) following a variant protocol for DNA isolation termed MiraPrep[64]. Following plasmid isolation, the eluate was precipitated by the addition of 3 M sodium acetate (Invitrogen, Waltham, MA, USA) at 10% (v/v) of eluate volume followed by the addition of 250% (v/v) absolute ethanol calculated after the addition of sodium acetate. The mixture was incubated at 4 °C overnight and centrifuged at 14,000 × *g* for 15 min at 4 °C. Supernatant was removed and 70% ethanol was added, followed by centrifugation at 14,000 × *g* for 10 min at 4 °C. Supernatant was removed once more, and precipitated DNA was resuspended in nuclease-free H$_2$O. Sequencing was conducted by Eurofins Genomics (Louisville, KY, USA), using standard sequencing primers provided by the company.

## Graphs and statistical analysis

Violin plots were made using the ggplot2 base package (version 3.1.0) as a component of the Tidyverse package (https://cran.r-project.org/web/packages/tidyverse/index.html) in rStudio (version 4.0.3). Wilcoxon ranked-sum tests to determine statistical significance between violin plots were conducted using base R statistics in rStudio. Recruitment plots, invasion assays, and all statistics associated with these data (pairwise T-test followed by Bonferroni post-analysis, SEM) were performed in Excel (Microsoft, Redmond, WA, USA). All graphs were assembled using the free and open-source software GNU Image

Manipulation Program (GIMP, https://www.gimp.org/) and Inkscape (https://inkscape.org/). Proposed model for Dyn2 oligomerization (Figs. 1–7, S4) was assembled using BioRender (https://app.biorender.com/).

## Reporting summary

Further information on research design is available in the Nature Portfolio Reporting Summary linked to this article.

## Data availability

Data supporting the findings of this manuscript are available within the article, Supplementary data, and within Raw Data files. Source data are provided with this paper and have been deposited to Dryad (https://doi.org/10.5061/dryad.brv15dvh5)[65]. Deposited raw data files include raw imaging data, source images for analysis, and spreadsheets containing analyzed data. Source data are provided with this paper.

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

## Acknowledgements
We thank the members of the Carabeo lab for extensive feedback regarding the design, direction, and analysis of the study. We thank Dr. Ken Fields (University of Kentucky College of Medicine) for the kind gift of ΔTarP, ΔTmeA, ΔTmeA/ΔTarP and TarP/TmeA *cis*-complemented strains. This study was supported by funding from the U.S. National Institutes of Health, National Institutes of Allergy and Infectious Disease grant R01 AIO65545 (R.A.C.) and by a fellowship from the Seattle Chapter of Achievement Rewards for College Scientists (M.D.R.). The contents and views expressed within this publication are the sole responsibility of the authors.

## Author contributions
Conceptualization: R.A.C.; Methodology: M.D.R.; Validation: M.D.R.; Formal analysis: M.D.R., R.A.C.; Investigation: M.D.R., R.A.C.; Resources: R.A.C.; Data curation: M.D.R.; Writing - original draft: M.D.R.; Writing - review & editing: R.A.C.; Visualization: M.D.R., R.A.C.; Supervision: R.A.C.; Project administration: R.A.C.; Funding acquisition: R.A.C.

## Competing interests
The authors declare no competing interests.
