## [Peer Review File · Nature Communications]

Dynamin-dependent entry of *Chlamydia trachomatis* is sequentially regulated by the effectors TarP and TmeAREVIEWER COMMENTS

Reviewer #1 (Remarks to the Author):

Entry of chlamydiae into epithelial cells requires rearrangements in actin architecture. This is orchestrated in part by the secretion of the effector proteins TarP and TmeA, and several studies already scrutinized their respective contribution to bacterial entry. Romeo and Carabeo implemented time-lapse videomicroscopy to record the kinetics of recruitment and turnover of proteins at the bacterial entry site. With this method, they showed that TarP accounted for most of the robust actin dynamics at invasion sites, with the recruitment of actin nucleators (formins and Arp2/3) (JCS 2022). Using the same approach, they focus in this paper on the kinetics of dynamin 2 (Dyn2) recruitment, and the relative contribution of TarP and TmeA in its behavior. Their data confirm that TarP and TmeA act in a sequential manner. They confirm the predominant role of TarP in orchestrating actin assembly and extend this finding to Dyn2, that co-distributes with actin. Time-lapse analyses allow to dissect which steps (Dyn2 recruitment, turnover, duration of the entry step) are affected when signaling pathways downstream of TarP (and already known for their effect on actin) are disturbed by chemical intervention (Fig. 3, 4). It is the first time that these signaling pathways are dissected at this time scale. Interestingly, a molecule that stimulates Dyn2 oligomerization, Ryngo1-23, restores Dyn2 recruitment profile, suggesting that TmeA helps Dyn2 oligomerization. A plausible scenario is proposed, in which TarP is implicated in Dyn2 recruitment while TmeA is involved later in Dyn2 oligomerization.

These data increase our understanding of Chlamydia orchestrate its own invasion. However, the intertwined signaling pathways and multiplicity of players limits solid conclusions, even with time lapse microscopy. The limitation of using chemical intervention is that the effect is not restricted to pathways immediately downstream of TarP. The proposed role of TmeA on Dyn2 oligomerization is a novelty but relies exclusively on the effect of the drug Ryngo 1-23. As for the recruitment of Dyn2 by TarP, it is not really a surprise as Dyn2 is well known to follows actin dynamics. The effect of TarP on Dyn2 distribution are consistent with Dyn2 recruitment being governed by TarP dependent actin recruitment.

Fig1 show that overexpression of a dominant negative form of Dyn2 delays the duration of internalization. This supports the implication of Dyn2 in efficient bacterial entry. The % infection efficiency in cells overexpressing Dyn2DN mutant is not shown. It is not needed, but since one of the claims of the paper is to revisit old conflicting results on the question of the implication of Dyn2 in Chlamydia entry, it would be nice to show the result. It would probably help the authors make the point that a significant difference in the rate of entry do not necessarily translate in a difference in entry efficiency, which has contributed to the confusion in the field. Data are always expressed as %MaxDyn2 MFI. Although it is useful to compare slopes, it does not allow to appreciate if there is a difference in overall recruitment of these molecules. The authors should also display Fold recruitment, as they did in the JCS 2022.

L161 incorrectly states that FigS2 shows that fluid phase uptake is Dyn2 independent. In fact, there is some agreement that micropinocytosis requires dynamin, while larger membrane reshuffling during macropinocytosis might not (PMID: 18003703). The TarP/TmeA independent entry of Chlamydia is not sufficiently well characterized to state that it is Dyn2 independent (see also below).

L165, based on data displayed in Fig. S2, the authors introduce a novel description of the pattern of Dyn2/actin recruitment that they observed upon infection in the TarP mutant. But they only show Dyn2 for the TarP mutant (not actin), making comparison with the TmeA mutant difficult. Also, in fig S2E, Actin should be in different color to make sure the reader can differentiate clearly Actin and Dyn2 distribution. In addition, before moving on with data presented in Fig3; I would expect a finer description of actin distribution upon infection with the TarP mutant. This question was studied in JCS 2022, however there was no mention in that paper of a change in actin distribution (focused versus more spread?), making this notion very confusing. Overall, the authors need to choose a better way to depict the distribution of actin/dyn2 in the

case of the mutants, because the phrasing so far does not fit with the images. For instance l.179: "prevented localized recruitment" does not fit with the data (I see Dyn2 enrichment in proximity of the bacteria), unless what is called "localized recruitment" is better defined. Similarly, l.209, the sentence suggests that the macropinocytic entry does not use Dyn2. This is not demonstrated in the paper, and the unfocused but clear enrichment of Dyn2 in the vicinity of the TarP mutants (Fig. S2 for instance) suggests the contrary.

At this point, to fully interpret the data, it seems necessary to determine what proportion of WT bacteria enter via the Tarp/TmeA promoted pathway, and what proportion uses macropinocytosis. The distinction on dextran co-localization (S2F) is very unconvincing, as the difference between the different strains are very small (which is not surprising, even TarP/TmeA assisted uptake is expected to take fluid phase) .

In Fig. 3, it is suggested that the effect of EHop-016 is mediated by TarP dependent activation of Rac1. To make this point, the authors need to show that EHop-016 has no effect on a TarP mutant. Same for wortmannin.

Dyn2 recruitment rate is higher in the TmeA mutant (Fig3, 5) but actin recruitment is not (Fig. 6). It seems relevant to discuss, since it would be the only time actin and Dyn2 dynamics diverge.

If TarP controls Dyn2 recruitment, we are not expecting Ryngo to restore Dyn2 "focused assembly", are we? Maybe a better quantification of this parameter will clarify this (S4).

As mentioned earlier, the restoration by Ryngo 1-23 of Dyn2 dynamics during invasion of the TmeA mutant is a very nice observation. It suggests role of TmeA in assisting Dyn2 oligomerization, however this is not corroborated by other kind of data (in vitro reconstitution for instance).

The authors made nice drawings to follow the steps at which each pharmaceutical intervention was expected to occur. A synthetic scheme placing the order of events controlled by TarP and TmeA would be welcome.

Reviewer #2 (Remarks to the Author):

This interesting manuscript is focused on the molecular mechanisms governing entry of *Chlamydia trachomatis* into epithelial cells. The chief innovation is the application of live-cell imaging to monitor events within the first seconds of chlamydial attachment. The authors leverage this approach to provide evidence for the involvement of host Dynamin2. Dynamin-mediated scission is responsive to actin polymerizing chlamydial effectors Tarp and TmeA. Of note, the narrow time-window allows the authors to resolve a controversy in whether Dynamin plays a role or not. They provide evidence that Dyn2 is required for WT rates of chlamydial entry. Dyn2 is not absolutely required for entry and this is likely where discrepancies occurred since other approaches focus on later time-points. The authors go on to link actin dynamics to Dyn2 recruitment and turnover and then implicate the effectors Tarp and TmeA, both of which stimulate actin polymerization, to the process. The strength of the study is that the level of resolution allows detection of differences among treatments. This leads to finding novel biology on both the host and bacterial side of the equation.

1. Dyn2 seems to play a minimal role—only delaying entry on the scale of minutes or slightly shifting recruitment rates. My biggest issue is that I think the authors are overselling the importance of their findings. For example, the first line of the discussion reads "In this study, we conclusively demonstrate that *C. trachomatis* utilizes host Dyn2 to complete invasion". This is an overstatement since the bacteria still invade. I think the authors can temper these statements by indicating a role of Dyn2 in one of many entry pathways for *Chlamydia*. This is alluded to several times, but it's never made clear that the pathway being addressed is one of multiple routes.
2. I'm a bit confused by the data in Fig 2C. How can the actin recruitment rates be different between dynasore and siRNA-treated cells if there is no role of Dyn2 in actin recruitment (no difference to mock or negative controls).

3. Did the authors try the tarp mutant complemented with a domain deletion abolishing the PI3K/Rac1 activating activity of Tarp? This would strengthen the data in figures 3 and 4 that relies on pharmacologic inhibitors.

4. How does one reconcile the conclusion that Tarp signaling via PI3K/Rac-1 governs Dyn2 with observations that only the C-terminal Tarp domain essential for direct actin polymerization is required for invasion (PMID 32152196)?

Minor comments

1. The authors may wish to use another word when describing host factors regulating dynamin oligomerization. The word "effector" is used (ie lines 57, 58) and this can cause confusion since type III secreted proteins are often referred to as "effectors".

2. There is an issue with the references section. For example, references 8 and 30 are the same.

3. Lines 271-273 indicate that neither Tarp nor TmeA directly interact with Dyn2. Has this been shown or is this an assumption?

Reviewer #3 (Remarks to the Author):

In the current manuscript the authors investigate the role of two chlamydial secreted effectors, TarP and TmeA and the host GTPase dynamin 2 in the invasion of *Chlamydia trachomatis* into host cells. The study is based on the use of chlamydial effector mutants and dynamin 2 chemical inhibitors/activators and knock down. The experiments suggest that TarP is involved in the recruitment of dynamin 2 whereas TmeA functions on the recruited dynamin 2 by inducing the oligomerization.

This is an elegant study mainly based on live-cell confocal microscopy to monitor chlamydial invasion and host factor recruitment and activation. The focus of the experiments is on the dynamics of events which are in the range of seconds for the recruitment and turnover studies and minutes for the invasion. The differential role of TarP and TmeA in dynamin 2 recruitment and oligomerization is nicely worked out now explaining conflicting results of some of the previous studies.

There are, however, two major concerns with this study.

Invasion of *C. trachomatis* is a multifactorial process involving several different receptors and probably also invasion mechanism. Deletion of TarP and TmeA reduces invasion efficiency by about 40%, in case of a double mutation by 60%. Since these events were measured at very early timepoints of infection at a seconds to minutes timescale for a narrow time window the overall infection and progeny formation is not monitored in these settings. Although it is not the focus of the study to investigate the infection outcome, it nevertheless is important to understand the overall significance of the findings.

The entire study has been performed in Cos7 cells for transfection experiments and HeLa for infection experiments (McCoy cells for propagation of *Chlamydia*). This means that even in one set of experiments cell lines from two different species, two different organs/tissues and two different cell types were used. Both are transformed cells which have a dysregulated cytoskeleton. The question is whether the mechanisms worked out here are relevant for invasion of *Chlamydia* into primary and polarized cells which are naturally invaded during infection. One should at least repeat some of the major experiments in cells that are non-transformed and have a native cytoskeleton.

Entry of chlamydiae into epithelial cells requires rearrangements in actin architecture. This is orchestrated in part by the secretion of the effector proteins TarP and TmeA, and several studies already scrutinized their respective contribution to bacterial entry. Romeo and Carabeo implemented time-lapse videomicroscopy to record the kinetics of recruitment and turnover of proteins at the bacterial entry site. With this method, they showed that TarP accounted for most of the robust actin dynamics at invasion sites, with the recruitment of actin nucleators (formins and Arp2/3) (JCS 2022). Using the same approach, they focus in this paper on the kinetics of dynamin 2 (Dyn2) recruitment, and the relative contribution of TarP and TmeA in its behavior. Their data confirm that TarP and TmeA act in a sequential manner. They confirm the predominant role of TarP in orchestrating actin assembly and extend this finding to Dyn2, that co-distributes with actin. Time-lapse analyses allow to dissect which steps (Dyn2 recruitment, turnover, duration of the entry step) are affected when signaling pathways downstream of TarP (and already known for their effect on actin) are disturbed by chemical intervention (Fig. 3, 4). It is the first time that these signaling pathways are dissected at this time scale. Interestingly, a molecule that stimulates Dyn2 oligomerization, Ryngo1-23, restores Dyn2 recruitment profile, suggesting that TmeA helps Dyn2 oligomerization. A plausible scenario is proposed, in which TarP is implicated in Dyn2 recruitment while TmeA is involved later in Dyn2 oligomerization.

These data increase our understanding of Chlamydia orchestrate its own invasion. However, the intertwined signaling pathways and multiplicity of players limits solid conclusions, even with time lapse microscopy. The limitation of using chemical intervention is that the effect is not restricted to pathways immediately downstream of TarP. The proposed role of TmeA on Dyn2 oligomerization is a novelty but relies exclusively on the effect of the drug Ryngo 1-23. As for the recruitment of Dyn2 by TarP, it is not really a surprise as Dyn2 is well known to follows actin dynamics. The effect of TarP on Dyn2 distribution are consistent with Dyn2 recruitment being governed by TarP dependent actin recruitment.

We thank Reviewer 1 for the thorough review. We agree that the intertwining of signaling pathways and the involvement of several participants complicate matters. Nevertheless, we were able to resolve an existing question in the field, which is dynamin involvement. Our conclusions that Dyn2 is involved is based on evaluating three parameters that we have previously shown to be important elements of the invasion process. They are recruitment/turnover profiles, invasion efficiency, and duration of internalization. To increase rigor, we modulated Dyn2 function via pharmacological inhibition, expression of dominant-negative mutants, and knockdown by siRNA. Collectively, data obtained point to a role of Dyn2 in invasion. Having implicated Dyn2 in invasion, the next logical question we addressed was its relationship to the two invasion-associated type III effectors, TarP and TmeA. In this regard, the manuscript identified recruitment to the invasion site to be mediated by TarP, followed by TmeA-dependent activation of Dyn2 to enable downstream oligomerization.

With regards to the use of Ryngo 1-23, we agree with the Reviewer that at this time, we cannot define its exact role in Dyn2 function. However, based on the ability of Ryngo to rescue the TmeA mutant, and its reported biochemical function related to Dyn2, we were able to

assemble a pathway consisting of the following steps: i) TarP-mediated Dyn2 recruitment, ii) TmeA-promoted post-recruitment event, and iii) functional activation of Dyn2. What the post-recruitment event could be, we could only speculate at this time. Having said this, we are nevertheless confident that it precedes oligomerization, which is a rate-limiting step in Dyn2 functionality. That the level of TarP-recruited Dyn2 in the absence of TmeA was not sufficient for oligomerization, suggests a TmeA-dependent process that primes oligomerization. This could be the promotion of formation of short oligomers as depicted in Figure 1B, or perhaps modification of the local microenvironment that facilitates oligomerization. Regardless, Ryngo 1-23 could bypass the uncharacterized TmeA-dependent process to rescue the TmeA mutant.

Fig1 show that overexpression of a dominant negative form of Dyn2 delays the duration of internalization. This supports the implication of Dyn2 in efficient bacterial entry. The % infection efficiency in cells overexpressing Dyn2DN mutant is not shown. It is not needed, but since one of the claims of the paper is to revisit old conflicting results on the question of the implication of Dyn2 in Chlamydia entry, it would be nice to show the result. It would probably help the authors make the point that a significant difference in the rate of entry do not necessarily translate in a difference in entry efficiency, which has contributed to the confusion in the field. Data are always expressed as %MaxDyn2 MFI. Although it is useful to compare slopes, it does not allow to appreciate if there is a difference in overall recruitment of these molecules. The authors should also display Fold recruitment, as they did in the JCS 2022.

We appreciate the Reviewer's point that demonstration of the effects of Dyn2DN on invasion efficiency and duration of internalization would better clarify past conflicting results. In Figure 1, we reframed the question of Dyn2 involvement in invasion from the point of view of assay sensitivity. Previous work by others quantified invasion efficiency through counting of inclusions at 5 h post-infection (Boleti et al, 1999.) or flow cytometry (Hybiske et al., 2007) as a function of Dyn2DN overexpression. Both approaches have their respective limitations. Specifically, inclusion development at 5 hpi is too far removed from invasion, which is completed in minutes. Quantifying invasion efficiency by flow cytometry limited observation at the population level. The purpose of Figure 1 is to present a new method of quantifying invasion at the single-EB level through measurement of duration of internalization that we felt would better evaluate Dyn2 involvement. This assay eliminates concerns associated with inclusion development as an indicator of invasion efficiency, as well as enabling single-EB analysis, which was not possible in the flow cytometry-based assay used by Hybiske et al. As shown in Figure 2, which utilized two different means of modulating Dyn2 function, e.g. Dynasore treatment and Dyn2 depletion by siRNA, duration of internalization is a suitable substitute to the traditional antibody-based assay of monitoring invasion. Having said this, the main point of Figure 2, was not to validate the new assay, but to demonstrate that invasion steps mediated by TarP and TmeA involved Dyn2. In the absence of either effector, Dyn2 inhibition did not produce further decrease in invasion efficiency.

L161 incorrectly states that FigS2 shows that fluid phase uptake is Dyn2 independent. In fact, there is some agreement that micropinocytosis requires dynamin, while larger membrane

reshuffling during macropinocytosis might not (PMID: 18003703). The TarP/TmeA independent entry of Chlamydia is not sufficiently well characterized to state that it is Dyn2 independent (see also below).

We appreciate bringing the indicated article to our attention and have amended the main body text to remove statements implying that fluid-phase uptake is de-facto dynamin independent (Line 163-164). Moreover, our determination that TarP/TmeA independent Chlamydia entry is dynamin-independent was on the basis that entry of these strains was insensitive to Dyn2 DN overexpression (Fig. S3), pharmacological inhibition of Dyn2 (Fig. 2G), and siRNA depletion of Dyn2 (Fig. 2H). As such, we are confident that entry of these strains is not reliant upon Dyn2 activity.

L165, based on data displayed in Fig. S2, the authors introduce a novel description of the pattern of Dyn2/actin recruitment that they observed upon infection in the TarP mutant. But they only show Dyn2 for the TarP mutant (not actin), making comparison with the TmeA mutant difficult. Also, in fig S2E, Actin should be in different color to make sure the reader can differentiate clearly Actin and Dyn2 distribution. In addition, before moving on with data presented in Fig3; I would expect a finer description of actin distribution upon infection with the TarP mutant. This question was studied in JCS 2022, however there was no mention in that paper of a change in actin distribution (focused versus more spread?), making this notion very confusing.

Overall, the authors need to choose a better way to depict the distribution of actin/dyn2 in the case of the mutants, because the phrasing so far does not fit with the images. For instance l.179: “prevented localized recruitment” does not fit with the data (I see Dyn2 enrichment in proximity of the bacteria), unless what is called “localized recruitment” is better defined. Similarly, l.209, the sentence suggests that the macropinocytic entry does not use Dyn2. This is not demonstrated in the paper, and the unfocused but clear enrichment of Dyn2 in the vicinity of the TarP mutants (Fig. S2 for instance) suggests the contrary.

We have incorporated this feedback into a revised version of Fig. S2, in particular panels S2E and S2F. Pseudocoloring was changed for these panels to more clearly differentiate actin and Dyn2 signal. Additionally, we reported on both Δ TarP (Fig. S2E) and Δ TmeA EBs (Fig. S2F), allowing readers to more easily compare the distribution of actin/Dyn2 recruitment exhibited by each strain. Descriptions of entry mechanisms or protein distribution in the JCS report were outside of the scope of that study, however we report that Δ TarP EBs elicited comparable actin distributions in both the JCS study and in the present study. Finally, we added a supplemental video comparing actin recruitment by wild-type, Δ TmeA, and Δ TarP EBs (Video S6). In this video, both wild-type and Δ TmeA EBs uniformly exhibit highly localized punctate recruitment of host proteins, whereas Δ TarP EBs uniformly exhibit unfocused or more diffused protein recruitment phenotypically similar to actin-rich ruffles or blooms. Aforementioned changes were reported in the body text (Lines 168, 171-173). We hope these changes improve clarity regarding our statements distinguishing protein distribution patterns.

In addition to the changes mentioned above (3a), we added a brief statement (Line 183-184) providing a rationale for our statement that “TarP deletion prevented localized recruitment of Dyn2” (Line 182). This distinction highlights that loss of TarP signaling prevents host protein recruitment at sites of EB contact, such that engulfment of Δ TarP EBs proceeds in a phenotypically and mechanistically distinct manner relative to wild-type and Δ TmeA EBs.

At this point, to fully interpret the data, it seems necessary to determine what proportion of WT bacteria enter via the Tarp/TmeA promoted pathway, and what proportion uses macropinocytosis. The distinction on dextran co-localization (S2F) is very unconvincing, as the difference between the different strains are very small (which is not surprising, even TarP/TmeA assisted uptake is expected to take fluid phase) .

The use of 10 kDa fluorescent dextran in this study was conducted on the basis that it serves as a general marker for multiple fluid-phase uptake mechanisms including, but not limited to, macropinocytosis (PMID: 25623938). In combination with a previous report indicating that invading wild-type EBs exclude dextran (PMID: 8641784), colocalization with 10 kDa dextran was intended to demonstrate the utilization of potentially non-canonical fluid-phase uptake mechanisms following TarP/TmeA deletion or disruption of host dynamins. Quantification of dextran-colocalized EBs was conducted in an unbiased manner using automated particle tracking across a Z-series to report on enrichment of fluorescent dextran signal coincident with fluorescent Chlamydia. As such, we are confident in the accuracy of our data regarding dextran colocalization and any reports in the manuscript referring to this data. To further support this claim, we additionally measured dextran colocalization rates following dynamin inhibition, noting that disruption of Dyn2 functionality by either inhibition (Dynasore) or hyperactivation (Ryngo 1-23) comparably increased the proportion of wild-type EBs that enter via fluid-phase. Consequently, we reiterate that Dyn2 activity contributes to canonical uptake of wild-type EBs, and that mutant strains more frequently utilize non-canonical entry mechanisms with features in common with fluid-phase uptake.

In Fig. 3, it is suggested that the effect of EHop-016 is mediated by TarP dependent activation of Rac1. To make this point, the authors need to show that EHop-016 has no effect on a TarP mutant. Same for wortmannin.

We appreciate this suggestion and agree that it would improve the rigor of our study. We conducted these experiments and reported the outcomes in a revised version of Fig. S4, specifically in panels S4B-E. We report that application of EHop-016 and Wortmannin have no observable effect on Δ TarP EB invasion and have amended the body text accordingly (Lines 193-195, 206-207).

Dyn2 recruitment rate is higher in the TmeA mutant (Fig3, 5) but actin recruitment is not (Fig. 6). It seems relevant to discuss, since it would be the only time actin and Dyn2 dynamics diverge.

Our previous study regarding the importance of TmeA signaling toward actin kinetics (PMID: 36093837) revealed that the TmeA mutant strain achieved slightly lower intensities of actin recruitment with slower recruitment kinetics owing to defective retention of Arp2/3. In contrast, Dyn2 recruitment intensity was unchanged by the TmeA mutant in this study (Fig. S5C) but achieved peak intensity at earlier timepoints following initiation of recruitment relative to wild-type Chlamydia. Actin recruitment activities associated with the TmeA signaling pathway more likely are distinct from the proposed TmeA-dependent Dyn2 oligomerization activity. Indeed, we determined that functional Dyn2 was associated with wild type rate of actin turnover. Loss of Dyn2 function, either through pharmacological inhibition or TmeA mutation led to abnormal turnover kinetic profiles. Ryngo treatment rescued defects in recruitment and turnover of both Dyn2 and actin in the TmeA mutant. With Ryngo previously characterized to promote Dyn2 oligomerization (PMID: 24891099), data for TmeA suggest that actin kinetics depend on Dyn2 oligomerization. It should be noted that in the TmeA mutant, the Dyn2 recruitment mechanism linked to TarP (Figs. 3,4) remain functional, which may account for the Dyn2 recruitment rate observed in the TmeA mutant.

If TarP controls Dyn2 recruitment, we are not expecting Ryngo to restore Dyn2 “focused assembly”, are we? Maybe a better quantification of this parameter will clarify this (S4).

The observation that Ryngo treatment prompted Dyn2 recruitment immediately at the host/EB interface of Δ TarP EBs (i.e., focused assembly) was unexpected, but ultimately did not yield rescue of TarP-independent entry by any observable metric. As such, we have omitted reference to this treatment within the body text to avoid confusion or implication of its significance.

As mentioned earlier, the restoration by Ryngo 1-23 of Dyn2 dynamics during invasion of the TmeA mutant is a very nice observation. It suggests role of TmeA in assisting Dyn2 oligomerization, however this is not corroborated by other kind of data (in vitro reconstitution for instance).

The Dyn2 oligomerization function of the small molecule compound Ryngo 1-23 is based on previous a previous report (PMID: 24891099); and our conclusion that assigned Dyn2 oligomerization to TmeA is based on this report in conjunction with Ryngo’s restoration of four key parameters – Dyn2 kinetics, actin kinetics, especially turnover, internalization duration, and partially, invasion efficiency exclusively in the TmeA mutant. We have included a statement to this effect in lines 235-239.

The authors made nice drawings to follow the steps at which each pharmaceutical intervention was expected to occur. A synthetic scheme placing the order of events controlled by TarP and TmeA would be welcome.

We appreciate this comment and agree that a model summarizing our findings would improve clarity regarding our investigation. As such, we generated a new figure (Fig. 7, Line

279) indicating the role of TarP and TmeA signaling in Dyn2-dependent entry of Chlamydia and highlighting the sequential nature of this interaction.

Reviewer #2 (Remarks to the Author):

This interesting manuscript is focused on the molecular mechanisms governing entry of Chlamydia trachomatis into epithelial cells. The chief innovation is the application of live-cell imaging to monitor events within the first seconds of chlamydial attachment. The authors leverage this approach to provide evidence for the involvement of host Dynamin2. Dynamin-mediated scission is responsive to actin polymerizing chlamydial effectors Tarp and TmeA. Of note, the narrow time-window allows the authors to resolve a controversy in whether Dynamin plays a role or not. They provide evidence that Dyn2 is required for WT rates of chlamydial entry. Dyn2 is not absolutely required for entry and this is likely where discrepancies occurred since other approaches focus on later time-points. The authors go on to link actin dynamics to Dyn2 recruitment and turnover and then implicate the effectors Tarp and TmeA, both of which stimulate actin polymerization, to the process. The strength of the study is that the level of resolution allows detection of differences among treatments. This leads to finding novel biology on both the host and bacterial side of the equation.

1. Dyn2 seems to play a minimal role—only delaying entry on the scale of minutes or slightly shifting recruitment rates. My biggest issue is that I think the authors are overselling the importance of their findings. For example, the first line of the discussion reads “In this study, we conclusively demonstrate that C. trachomatis utilizes host Dyn2 to complete invasion”. This is an overstatement since the bacteria still invade. I think the authors can temper these statements by indicating a role of Dyn2 in one of many entry pathways for Chlamydia. This is alluded to several times, but it’s never made clear that the pathway being addressed is one of multiple routes.

Although our data consistently demonstrate that Dyn2 activity is necessary for optimal entry, and that defects associated with TmeA deletion are strongly correlated with defective utilization of a Dyn2-mediated entry mechanism, we also acknowledge that Dyn2 disruption achieved only partial impairment of invasion in wild-type Chlamydiae. To address this, we added further information about the plurality of entry mechanisms used by Chlamydia (Lines 46-48, 163-164), and included a statement acknowledging the partial phenotype described in this study (Lines 309-313).

2. I’m a bit confused by the data in Fig 2C. How can the actin recruitment rates be different between dynasore and siRNA-treated cells if there is no role of Dyn2 in actin recruitment (no difference to mock or negative controls).

Our statement in lines 144-145 “Interestingly, actin recruitment kinetics were largely unchanged by Dyn2 disruption...” was made to underscore that 3 out of 4 groups (Mock, Scramble, Dyn2 siRNA) exhibited no observable defect in actin recruitment, whereas

Dynasore treatment had only a slight (~20%) reduction in actin recruitment rate. Given that actin recruitment rates are variable in both mock (+/- 50%) and Dynasore (+/- 40%) treated groups, significance via Wilcoxon rank-sum was not achieved. Significance between Dynasore and Dyn2 siRNA treated groups in Fig. 2C may be due to partial (50%) knockdown of Dyn2 and/or tighter overall distribution of rates in the latter group. We opted not to discuss this in the main body text given that the difference was marginal (-20%, only in Dynasore-treated cells) and inconsistent (no change in Dyn2 siRNA-treated cells).

3. Did the authors try the tarp mutant complemented with a domain deletion abolishing the PI3K/Rac1 activating activity of Tarp? This would strengthen the data in figures 3 and 4 that relies on pharmacologic inhibitors.

We agree that mutant complementation with domain-deletion variants of TarP and TmeA are extremely useful tools for investigating the regulatory interactions that underpin Dyn2 utilization. Presently, we are underway in developing and employing these tools to generate a comprehensive model which integrates the signaling contributions of each effector toward Dyn2 recruitment and activation (Lines 311-313). We believe that this line of inquiry is more suitable for a follow-up investigation that focuses on TarP.

4. How does one reconcile the conclusion that Tarp signaling via PI3K/Rac-1 governs Dyn2 with observations that only the C-terminal Tarp domain essential for direct actin polymerization is required for invasion (PMID 32152196)?

We acknowledge the importance of the C-terminal domains of C. trachomatis TarP toward its role in invasion but contend that the identity and function of these domains are in dispute. Previously, our group has published that the aforementioned F-actin binding domains are instead a FAK-binding domain (PMID: 25193659) and two tandem vinculin-binding domains (PMID: 26649283). In further support of this, a comprehensive yeast-two-hybrid screen conducted in 2022 to characterize the interactome between TarP and host proteins yielded only one bona-fide interaction, specifically between vinculin and TarP (PMID: 36093837). We speculate that the interaction between vinculin and TarP is crucial to invasion and are presently investigating.

Minor comments

1. The authors may wish to use another word when describing host factors regulating dynamin oligomerization. The word “effector” is used (ie lines 57, 58) and this can cause confusion since type III secreted proteins are often referred to as “effectors”.

Instances where “effector” is used to describe host regulators of Dyn2 oligomerization (Lines 59, 60) have been changed to “activator” to avoid confusion.

2. There is an issue with the references section. For example, references 8 and 30 are the same.

Thank you for spotting this error, we have amended any inadvertent duplications of references in the body text and ensured that any sources referred to multiple times in the study all use the same reference number in the references section.

3. Lines 271-273 indicate that neither Tarp nor TmeA directly interact with Dyn2. Has this been shown or is this an assumption?

We have conducted proteomics screens of TarP using both yeast-two-hybrid (PMID: 36093837) and APEX2 proximity proteomics (unpublished), wherein neither screen yielded Dyn2 as TarP-interacting protein. Likewise, the TmeA-APEX2 proximity proteomics screen conducted by Keb et. al (PMID: 33468693) did not identify Dyn2 as a TmeA-interacting protein. On this basis, we contend that Dyn2 does not directly interact with either effector.

Reviewer #3 (Remarks to the Author):

In the current manuscript the authors investigate the role of two chlamydial secreted effectors, TarP and TmeA and the host GTPase dynamin 2 in the invasion of Chlamydia trachomatis into host cells. The study is based on the use of chlamydial effector mutants and dynamin 2 chemical inhibitors/activators and knock down. The experiments suggest that TarP is involved in the recruitment of dynamin 2 whereas TmeA functions on the recruited dynamin 2 by inducing the oligomerization.

This is an elegant study mainly based on live-cell confocal microscopy to monitor chlamydial invasion and host factor recruitment and activation. The focus of the experiments is on the dynamics of events which are in the range of seconds for the recruitment and turnover studies and minutes for the invasion. The differential role of TarP and TmeA in dynamin 2 recruitment and oligomerization is nicely worked out now explaining conflicting results of some of the previous studies.

There are, however, two major concerns with this study.

Invasion of C. trachomatis is a multifactorial process involving several different receptors and probably also invasion mechanism. Deletion of Tarp and TmeA reduces invasion efficiency by about 40%, in case of a double mutation by 60%. Since these events were measured at very early timepoints of infection at a seconds to minutes timescale for a narrow time window the overall infection and progeny formation is not monitored in these settings. Although it is not the focus of the study to investigate the infection outcome, it nevertheless is important to understand the overall significance of the findings.

Titers obtained from culture and purification of wild-type, Δ TarP, Δ TmeA, and Δ TmeA/ Δ TarP double knockout strains are as follows: Wild-type = 7.1×10^9 IFU/mL, Δ TarP = 5.4×10^9 IFU/mL, Δ TmeA = 6.2×10^9 IFU/mL, Δ TmeA/ Δ TarP = 0.4×10^9 IFU/mL. In agreement with

these data, we note that Δ TarP and Δ TmeA mutant strains did not exhibit defects in developmental cycle progression or IFU recovery. Meanwhile the Δ TmeA/ Δ TarP double mutant exhibited poor IFU recovery, suggesting the presence of a developmental defect in this strain. Altogether, our observations are consistent with previous reports by Keb et. al (PMID: 33468693) wherein the double knockout strain was reported to experience a defect in early development.

For our invasion assays and dextran uptake assays, a timepoint of 10 minutes post-inoculation was selected to optimize for invasion-competent EB uptake (i.e., pathogen-directed EB uptake). Later timepoints are likely to include invasion-incompetent EBs whose uptake occurs via host-directed nonspecific entry mechanisms. Data obtained from live-cell imaging exemplify this phenomenon; footage obtained between 0-30 minutes post-inoculation (Video S6) depict numerous entry events downstream of pathogen-directed induction of signaling. In contrast, videos monitoring invasion outside this window (i.e., starting at 30 min post-inoculation, Video S7) exhibit numerous invasion-incompetent EBs or particles with obvious defects in recruitment of specific proteins, such as Dyn2 and actin.

Finally, we are actively investigating the immediate intracellular fate (i.e., trafficking) of EBs that are internalized following dynamin inhibition to determine if upstream perturbation of dynamin-dependent entry induces downstream defects in intracellular trafficking, tracking their immediate intracellular fates.

The entire study has been performed in Cos7 cells for transfection experiments and HeLa for infection experiments (McCoy cells for propagation of Chlamydia). This means that even in one set of experiments cell lines from two different species, two different organs/tissues and two different cell types were used. Both are transformed cells which have a dysregulated cytoskeleton. The question is whether the mechanisms worked out here are relevant for invasion of Chlamydia into primary and polarized cells which are naturally invaded during infection. One should at least repeat some of the major experiments in cells that are non-transformed and have a native cytoskeleton.

We acknowledge that the cell lines used for the bulk of this study are not the native human host for C. trachomatis infection, raising potential concerns about the applicability of this study toward human infection. As such, we have repeated key experiments in this study (e.g., invasion assay, Dyn2/actin recruitment) using primary human cervical epithelial (HCE) cells and reported our findings in Fig. S6. We report that invasion efficiency data in both HeLa and HCE cell lineages are directly comparable (Fig. S6A-D). Since HCE cells poorly tolerate transfection and possess a cellular architecture, (i.e. domed rather than flat), unsuitable for quantitative imaging, we were limited to fixed-cell immunofluorescence to quantify recruitment of Dyn2 and actin (Fig. S6E-J). While we were unable to obtain kinetics data, we were able to implicate Dyn2 in invasion. We report that disruption of Dyn2 activity (i.e., Dynasore or Ryngo 1-23 treatment) altered Dyn2 and actin recruitment for wild-type EBs (Fig. S6E,F). In contrast, Δ TarP and Δ TmeA EBs were insensitive to Dynasore treatment, while Ryngo 1-23 administration increased the quantity of Dyn2⁺ and actin⁺ colocalization events exclusively for Δ TmeA EBs (Fig. S6G-J). Altogether, we conclude that similar trends associated with dynamin disruption in Cos7 cells are also present in primary HCE cells. Collectively, our

data regarding invasion efficiency and Dyn2/actin recruitment using HCE cells indicate that the mechanism described in our study is also present in the native host epithelial targets of C. trachomatis infection. In addition to generating a new supplemental figure reporting the aforementioned findings (Fig. S6), we updated the main body text to refer to these findings (Lines 272-275).

REVIEWER COMMENTS

Reviewer #1 (Remarks to the Author):

The authors have addressed my main concerns except for the need to back the hypothesis of TmeA's role on Dyn2 polymerization by biochemical observations. The legend of figure 7 should acknowledge that the role of TmeA on Dyn2 oligomerization is a working hypothesis at this stage. Also, in that figure the arrow linking Tarp to Dyn2 recruitment suggests a direct link, which is misleading. I would move this arrow to bridge Actin+ "effector" and Dyn2 recruitment (in other words, there is no evidence for Dyn2 recruitment by TarP other than via the recruitment of actin and accessory molecules, which themselves recruit Dyn2. The word "effector" should be avoided as TarP and TmeA are called effectors).

L259: "and DTmeA" is missing

Reviewer #2 (Remarks to the Author):

Overall, the authors have adequately responded to my previous concerns. The exception is point 4. I agree with the authors that the identity of host proteins binding the C-terminus of Tarp is subject to interpretation. However, the proposed model links Tarp activation of Rac1 with recruitment of Dyn2. This event is important for invasion. This is reported to require the tyrosine repeat domain of Tarp. My point was that genetic studies indicate that the C-terminus, but not the repeat domain, are necessary for the invasion phenotype. I'm scratching my head trying to figure out how this makes sense. The original report linking Tarp to Rac1 was done before the benefit of definitive genetics. Has it been confirmed that loss of Tarp results in the loss of Rac1 activation during chlamydial entry?

Reviewer #3 (Remarks to the Author):

The authors conducted additional experiments and repeated the inhibitor study in primary human cervical cells. Some of the most important results are thus confirmed in a relevant cell model. The authors have therefore addressed my points appropriately in the revised version of the manuscript.

REVIEWER COMMENTS

Reviewer #1 (Remarks to the Author):

The authors have addressed my main concerns except for the need to back the hypothesis of TmeA's role on Dyn2 polymerization by biochemical observations. The legend of figure 7 should acknowledge that the role of TmeA on Dyn2 oligomerization is a working hypothesis at this stage. Also, in that figure the arrow linking TarP to Dyn2 recruitment suggests a direct link, which is misleading. I would move this arrow to bridge Actin+ "effector" and Dyn2 recruitment (in other words, there is no evidence for Dyn2 recruitment by TarP other than via the recruitment of actin and accessory molecules, which themselves recruit Dyn2. The word "effector" should be avoided as TarP and TmeA are called effectors).

We thank the Reviewer for their thorough review.

The requested change has been made (Lines 858-860). Figure 7 legend now reads as follows with the modified passage highlighted below:

(A) TarP signaling promotes Rac1 activation downstream of PI3K signaling within the tandem N-terminal phosphodomains of the effector. Subsequent activation of the Arp2/3 complex enhances actin polymerization and promotes Dyn2 accumulation, providing a basis for TarP-mediated regulation of Dyn2 recruitment that is sensitive to inhibition of PI3K (Wortmannin) or Rac1 (EHop-016). Rescue of TmeA deficiency by the dynamin-activating compound Ryngo 1-23 is consistent with a working model whereby TmeA mediates oligomerization and subsequent assembly of scission-competent Dyn2 helical collars. (B) Signaling for TarP and TmeA promotes dynamin-dependent entry of Chlamydia in a sequential and synergetic manner, such that TarP signaling regulates Dyn2 recruitment whereas TmeA signaling regulates Dyn2 polymerization.

Regarding Figure 7B, the arrow has been changed to indicate that TarP's function in mediating recruitment of Dyn2 is indirect.

As for the use of the term "effector", we have changed it to "accessories".

L259: "and DTmeA" is missing

"and Δ TmeA" has been added in Line 263 (new numbering).

Reviewer #2 (Remarks to the Author):

Overall, the authors have adequately responded to my previous concerns. The exception is point 4. I agree with the authors that the identity of host proteins binding the C-terminus of TarP is subject to interpretation. However, the proposed model links TarP activation of Rac1 with recruitment of Dyn2. This event is important for invasion. This is reported to require the tyrosine repeat domain of TarP. My point was that genetic studies indicate that the C-terminus, but not the repeat domain, are necessary for the invasion phenotype. I'm scratching my head trying to figure out how this makes sense. The original

report linking Tarp to Rac1 was done before the benefit of definitive genetics. Has it been confirmed that loss of Tarp results in the loss of Rac1 activation during chlamydial entry?

We thank the Reviewer for this thoughtful comment. Rac1 activation during invasion has not been assessed in the Δ tarP mutants. However, we and others have obtained several lines of evidence that show Rac1 signaling to be associated with TarP. The TarP dependence of Rac1's involvement in invasion has been shown biochemically (PMID: 18383626) and genetically (PMID: 33468693). The former demonstrated through pulldown assays interaction of Rac1 to TarP via the Sos1/Abi1/Eps8 complex in a manner that required TarP phosphorylation at a specific tyrosine residue within a single copy of the repeated phosphodomain. The latter utilized the Rac1 inhibitor Ehop-016 (25 μ M) in conjunction with the Δ tarP mutant strain, for which they found no further decrease in invasion. This was interpreted to mean that Rac1 and TarP functioned in the same signaling pathway. In contrast, Ehop-016 treatment affected invasion of the Δ tmeA mutant strain, indicating that Rac function is not part of TmeA signaling. We were able to confirm this finding in this manuscript (Supplemental figure 6, panels A and C; Δ tarP vs. Δ tarP+Ehop-016). Data were from three independent replicates, but were split into different treatment groups in the figure for clarity. The same group evaluated invasion efficiencies of Δ tarP mutants complemented with different TarP deletion derivatives to determine TarP domain requirements. While the C-terminal domain of TarP, specifically the F-actin binding domain was found to be involved ($p < 0.001$), the phosphodomain was found to be necessary ($p < 0.05$) (PMID: 32152196, Figure 4), in spite of its deemphasis in the narrative.

We also note that invasion was assessed at 60 min post-inoculation, which was significantly longer than what we used (10 min), which could account for the reported diminished role of the phosphodomain. The extended invasion protocol by Jewett et al. likely masked the role of the phosphodomain, and by extension, of Rac1 by enabling alternative means of EB internalization. Our choice of the shorter duration was based on previous findings from single-EB analysis in quantitative live-cell imaging experiments (most recently in PMID: 36093837) that 10 min was sufficient to capture actin recruitment, turnover, and EB internalization events. With regards to Rac1 inhibition, we cannot conclude at this time if inefficient invasion is due to inefficient recruitment of actin, Dyn2, or both. We suspect that the two events are intertwined given the several reports on interaction of Dyn2 with actin. This is reflected in our working model shown in Figure 7.

For clarification, we have included the following passage (Lines 200-204) to highlight the role of the phosphodomain in invasion, which would support our observations from Rac1 inhibition experiments. We also added a qualifying statement that considers the Dyn2- and actin-related roles of TarP-Rac1 signaling axis.

*“A previous report that took advantage of allelic exchange mutagenesis to create *C. trachomatis* Δ tarP mutant strains complemented with TarP versions lacking specific domains implicated the phosphodomain in invasion³⁶. This observation is consistent with a role for Rac, which signals from the phosphodomain. At this time, we cannot conclude if inefficient invasion upon Rac inhibition is due to aberrant Dyn2 or actin dynamics at invasion sites, or both.”*

Reviewer #3 (Remarks to the Author):

The authors conducted additional experiments and repeated the inhibitor study in primary human cervical cells. Some of the most important results are thus confirmed in a relevant cell model. The authors have therefore addressed my points appropriately in the revised version of the manuscript.

We thank the Reviewer for the thorough and helpful critique of the manuscript.